

# Photo-initiated ground state chemistry: How important is it in the atmosphere?

Keiran N. Rowell[1,2], Scott H. Kable[2], and Meredith J. T. Jordan[1]

[1]School of Chemistry, University of Sydney, Sydney, Australia
[2]School of Chemistry, University of New South Wales, Sydney, Australia

**Correspondence:** Meredith Jordan (meredith.jordan@sydney.edu.au)

**Abstract.** Carbonyls are among the most abundant volatile organic compounds in the atmosphere. They are central to atmospheric photochemistry as absorption of near-UV radiation by the C=O chromophore can lead to photolysis. If photolysis does not occur on electronic excited states, non-radiative relaxation to the ground state will form carbonyls with extremely high internal energy. These 'hot' molecules can access a range of ground state reactions. Up to nine potential ground state reactions
are investigated at the B2GP-PLYP-D3/def2-TZVP level of theory for a dataset of 20 representative carbonyls. Almost all are energetically accessible under tropospheric conditions. Comparison with experiment suggests the most significant ground state dissociation pathways will be concerted triple fragmentation in saturated aldehydes, Norrish type III dissociation to form another carbonyl, and $H_2$–loss involving the formyl H atom in aldehydes. Tautomerisation, leading to more reactive unsaturated species, is also predicted to be energetically accessible and is likely to be important when there is no low-energy ground state
dissociation pathway, for example in $\alpha,\beta$-unsaturated carbonyls and some ketones. The concerted triple fragmentation and $H_2$–loss pathways have immediate atmospheric implication to global $H_2$ production and tautomerisaton has implication to the atmospheric production of organic acids.

## 1   Introduction

Carbonyls are a class of volatile organic compounds (VOCs) central to atmospheric chemistry. They arise, in large quantities,
from primary anthropogenic and biogenic emissions and via secondary atmospheric processes (Kesselmeier and Staudt, 1999; Millet et al., 2010; Chen et al., 2014). For example, small carbonyls, which rank at the top of anthropogenic emissions (Simon et al., 2010), are emitted as pollutants (Chen et al., 2014) and up to 10% of carbon initially fixed by plants is subsequently emitted as biological volatile organic compounds (BVOCs) that oxidise to carbonyls (Seco et al., 2007). Indeed, carbonyls are generated throughout the oxidative degradation pathways of all VOCs (Carlier et al., 1986; Atkinson and Arey, 2003).
Atmospheric concentrations of carbonyls are in the pptV-ppbV range (Tanner et al., 1996), with high concentrations found at low altitudes and in polluted environments (Lee et al., 1998; Pal et al., 2008; Guo et al., 2014; Menchaca-Torre et al., 2015).

As there are of the order of $10^6$ known or suspected VOCs in the atmosphere (Goldstein and Galbally, 2007), with major BVOCs being typically structurally complex, atmospheric carbonyls can have varied chemical structures and reactivities (Kawamura et al., 2000; Atkinson et al., 2006). Structural complexities include double-bonds, branched carbon chains and



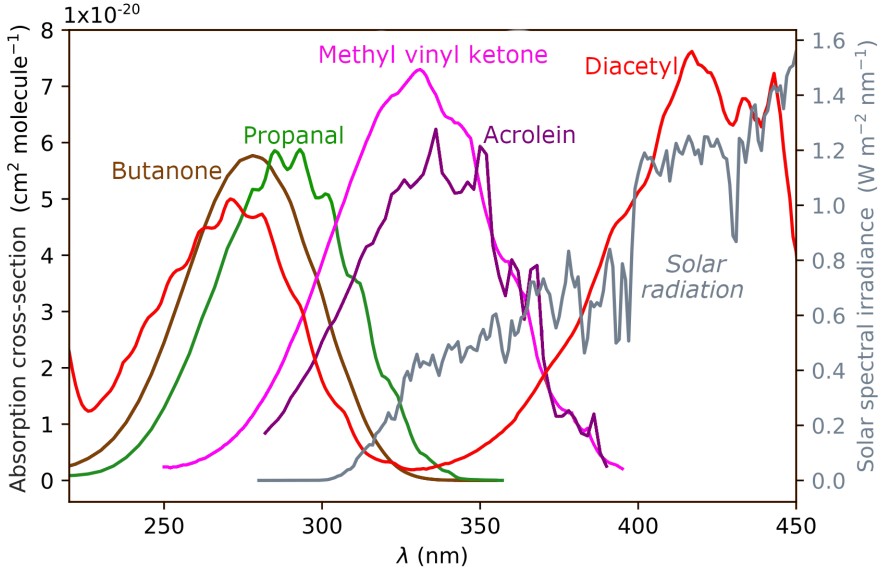

**Figure 1.** UV absorption spectra of representative carbonyls: aldehydes (represented by propanal), ketones (butanone), enals (acrolein), enones (methyl vinyl ketone), and $\alpha$-dicarbonyls (diacetyl). The solar spectrum is shown in grey, plotted against the right ordinate.

varied functionalisation. The atmospheric importance of carbonyls arises because they are one of the few classes of VOC that can absorb solar radiation in the troposphere. In the atmosphere, carbonyl photochemistry is most notable for its generation of radicals following excited state bond cleavage, the Norrish Type I reaction (NTI) (Norrish and Appleyard, 1934), and these radicals drive key atmospheric reactions (Lary and Shallcross, 2000; Yang et al., 2018). However, many other reactions, including reactions on the ground electronic state, $S_0$, are possible.

The intensity of solar radiation reaching the troposphere is shown in gray on the right axis of Fig. 1. Most UV-B radiation (280–315 nm) is absorbed by the ozone layer, and the intensity of UV-A radiation (315-400 nm) increases with increasing wavelength. The maximum photon energy available in the troposphere is $\sim$400 kJ/mol (300 nm) and is shown in violet on relevant energetic plots below. The overlap of a given carbonyl absorption spectrum and the solar spectrum defines the actinic energy window in which photo-initiated chemistry can occur. In carbonyls, a near-UV photon excites the $S_1 \leftarrow S_0$ $(n, \pi^*)$ tran-

sition of the C=O chromophore. The absorption wavelength depends on the structural class of the carbonyl and is determined by the substituents attached to the C=O. UV absorption spectra for representative carbonyls are shown in Fig. 1.

    Over the past couple of decades, it has become apparent that there are carbonyl "photochemical" pathways on the ground electronic state ($S_0$). These photo-initiated reactions on $S_0$ include the novel class of 'roaming' reactions (Townsend et al., 2004; Bowman and Houston, 2017), where partly dissociated radical fragments become trapped in each other's van der Waals

well and recombine to generate molecular products. However, more conventional $S_0$ transition state (TS) reactions, which occur over a barrier, as well as barrierless $S_0$ reactions, are also being discovered (Heazlewood et al., 2011; So et al., 2018; Toulson



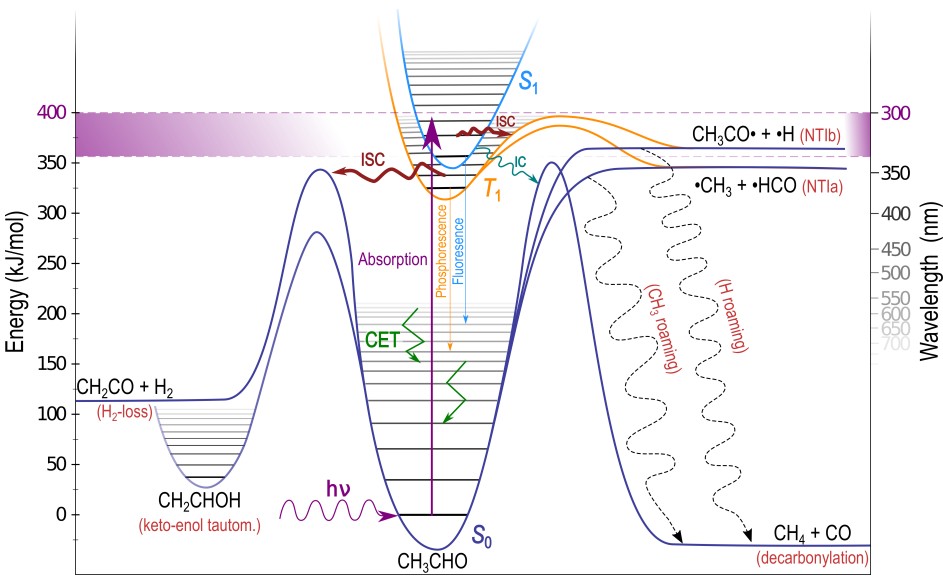

**Figure 2.** Schematic of the atmospheric fate of acetaldehyde ($CH_3CHO$) following absorption of a near-UV photon. Possible outcomes depend on excitation energy and include: electronic surface crossing, collisional energy transfer (CET), unimolecular dissociation including roaming (dashed arrows), and isomerisation.

et al., 2018). Despite the growing evidence of carbonyl ground state reactions, following excitation at energies accessible in the troposphere ($\lambda > 300$ nm), these reactions are almost entirely absent from the reaction schemes of contemporary atmospheric models (Wild et al., 2000; Rickard and Young; Jenkin et al., 1997, 2002; Harvard Atmospheric Chemistry Modeling Group).

## 1.1 The Tropospheric Fate of Carbonyls

The tropospheric fate of an illustrative carbonyl, acetaldehyde ($CH_3CHO$) is shown schematically in Fig. 2. A near-UV photon excites the $S_1 \leftarrow S_0$ ($n, \pi^*$) transition. The actinic absorption window, indicated in Fig. 2 by violet shading, is bounded by the lowest energy excitation to $S_1$ and the highest energy photon available in the troposphere. For most carbonyls the $S_1$ state is bound at actinic energies, as shown in Fig. 2, and $S_1$ reactions are inaccessible (Haas, 2004; Rowell et al., 2019). Intersystem crossing (ISC) from $S_1$ to the lowest energy triplet state, $T_1$, is formally spin-forbidden, however, as shown schematically in Fig. 2, carbonyl $S_1$ and $T_1$ minimum energy geometries are almost identical and the energetic separation between them is small. This allows relatively fast $S_1 \rightarrow T_1$ ISC (Hansen and Lee, 1975), indicated in Fig. 2 by a thick ISC arrow. On $T_1$, NTI $\alpha$-bond cleavage is the most common photolysis reaction (Kirkbride and Norrish, 1931; Zhu et al., 2009) and we define NTI$a$ and NTI$b$ as cleavage of the $\alpha$-bond to the larger and smaller substituents, respectively. In $CH_3CHO$ NTI$a$ results in formation of $^\bullet CH_3 + {}^\bullet HCO$ and NTI$b$ yields $CH_3CO^\bullet + {}^\bullet H$. On $T_1$ both NTI$a$ and NTI$b$ involve an energetic barrier, shown in orange in Fig. 2.



For larger carbonyls, the Norrish Type II (NTII) intramolecular hydrogen-transfer reaction, forming an alkene and an enol, also occurs and is initiated in an electronic excited state (Tadić et al., 2002, 2012). Although NTI and NTII are typically the major contributors to the photolysis quantum yield (QY) of carbonyls (Wagner and Zepp, 1972; Wagner and Klán, 2004; Zhu et al., 2009), if excitation is to energies below excited state photolysis thresholds, or if the electronically excited carbonyl is collisionally cooled below these thresholds, non-radiative relaxation to $S_0$ must occur; fluorescence and phosphorescence QYs in carbonyls are typically <1% (Heicklen and Noyes, 1959; Copeland and Crosley, 1985). The large energy separation between $S_1$ and $S_0$ means $S_1 \rightarrow S_0$ internal conversion (IC) will be slow and less important than ISC (Lee and Chen, 1997). Indeed, rapid IC is usually associated with conical intersections: geometric configurations where two electronic surfaces are degenerate (Schuurman and Stolow, 2018). In the main, these lie above the actinic energy range in carbonyls (Diau et al., 2001; Chen and Fang, 2009; Toulson et al., 2017).

The carbonyl RC=O group is planar in the $S_0$ minimum energy geometry but pyramidalised in the $S_1$ and $T_1$ minima (Godunov et al., 1995). The change in electron spin, geometric dissimilarity and energetic separation between the $T_1$ and $S_0$ minima suggest $T_1 \rightarrow S_0$ ISC may be slow. However, the non-bonding oxygen lone pair orbital, $n$, in C=O is approximately orthogonal to the antibonding $\pi^*$ excited state orbital. Thus the change in electron spin angular momentum in $T_1 \rightarrow S_0$ relaxation is accompanied by a change in electron orbital angular momentum. This results in conservation of total electron angular momentum and a high $T_1 \rightarrow S_0$ ISC rate, as dictated by El-Sayed's rule (El-Sayed, 1961). Indeed, ISC to $S_0$ can be kinetically competitive even when NTI reactions are accessible (Heazlewood et al., 2009; Amaral et al., 2010).

The most likely carbonyl relaxation route to $S_0$ is therefore via two sequential ISC steps (Toulson et al., 2017). This generates vibrationally 'hot' photo-excited $S_0$ molecules that retain much of the initial UV photon energy as internal energy. This 'hot' carbonyl can collide with other atmospheric molecules and collisional energy transfer (CET) removes excess vibrational energy, returning the molecule to thermal equilibrium. This is indicated in Fig. 2 by angular green arrows in the $S_0$ well, although CET will also occur in $S_1$, $T_1$ and isomer wells (Heazlewood et al., 2011; Andrews et al., 2012). The photo-excited $S_0$ carbonyls are conformationally flexible, and $S_0$ dissociation and isomerisation will compete with CET.

This paper explores, using computational chemistry, the possible $S_0$ dissociation and isomerisation pathways and how they depend on the structural class of the carbonyl.

## 1.2 The Dataset

There are a vast number of carbonyls in the atmosphere with little experimental data in comparison to this diversity (Atkinson et al., 1992). Here we consider a dataset of 20 carbonyls, which includes the 12 species that have explicit photolysis reactions within the Master Chemical Mechanism (Jenkin et al., 1997; Rickard and Young) and five species in the GEOS-Chem (Harvard Atmospheric Chemistry Modeling Group) atmospheric chemistry models. We also include carbonyls with atmospherically representative structural features based on bond order or substitution around the C=O chromophore. The 20 carbonyls in the dataset are shown in Fig. 3 and were previously used in a study of excited state NTI photolysis (Rowell et al., 2019). They are colour-coded into seven structurally distinct classes, which determine their absorption spectra (Fig. 1) and chemical behaviour (Atkinson et al., 1992).





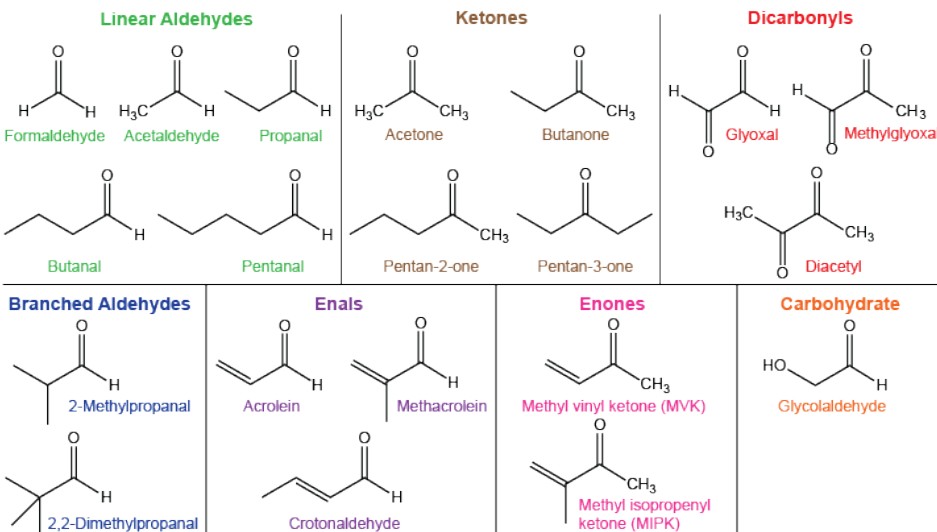

**Figure 3.** The 20 carbonyls in the dataset, colour-coded according to carbonyl class.

Quantum chemistry methods are used to calculate reaction thresholds for up to nine $S_0$ unimolecular reactions that may be accessible under tropospheric conditions. In the figures below, these are colour-coded according to Scheme 3. Explicitly, the reactions are:

- Decarbonylation (CO–loss).

- Concerted triple fragmentation (TF).

- Norrish Type III $\beta$-H transfer (NTIII).

- $H_2$–loss from hydrogens at the formyl and $\alpha$ positions (formyl+$\alpha$).

- $H_2$–loss from hydrogens at the $\alpha$ and $\beta$ positions ($\alpha$+$\beta$).

- $H_2$–loss from hydrogens at the $\beta$ and $\gamma$ positions ($\beta$+$\gamma$).

- Alkane/alkene elimination (AE) in saturated/unsaturated ketones.

- Keto–enol tautomerisation.

- Enal–ketene tautomerisation.

Ground state NTI reactions are also possible for all carbonyls in the dataset. These reactions are barrierless and their asymptotic energies have been previously reported (Rowell et al., 2019). Other isomerisations are possible. For example, acetaldehyde can
isomerise to oxirane or methylhydroxycarbene. Although these isomers are theoretically accessible at actinic energies, their formation barriers are very high with low barriers for the reverse isomerisation. Thus they are unlikely to be collisionally





stabilised before isomerising back to the parent carbonyl (Heazlewood et al., 2011). Norrish type II reactions are possible for butanal, pentanal and pentan-2-one and have not been considered here as they involve excited electronic states (Tadić et al., 2002, 2012).

Seven of the $S_0$ reactions listed above are illustrated for butanal in Fig. S1 of the Supplementary Material; alkane elimination is only available to ketones and enal–ketene tautomerisation is only available to acrolein, methacrolein and crotonaldehyde. The figure also includes schematic representations of the concerted TF and NTIII mechanisms.

The calculated $S_0$ thresholds are used to determine general trends that can be applied to larger carbonyls and to identify the most likely $S_0$ reactions for each class of carbonyl under tropospheric conditions. These reactions are then assessed in terms
of their tropospheric significance.

## 2   Computational Methods

$S_0$ calculations were performed with the B2GP-PLYP double-hybrid density functional (Karton et al., 2008), using the def2-TZVP canonical basis set (Weigend and Ahlrichs, 2005) and the RIJK resolution of the identity approximation with the def2/JK and def2-TZVP/C auxiliary basis sets. The use of the RIJK approximation, and the 'RI-' prefix, are taken as implicit. All
calculations were dispersion-corrected using the D3(BJ) scheme (Grimme et al., 2010, 2011), abbreviated D3 below. Geometry optimisations and frequency calculations were performed using the ORCA electronic structure program (Neese, 2017). All zero-point energies were scaled with the literature scaling factor of 0.9752 (Kesharwani et al., 2015).

The B2GP-PLYP functional has been found to give errors <8 kJ/mol across a diverse range of benchmark data (Karton et al., 2008). B2GP-PLYP-D3/def2-TZVP was also validated by the authors against experimental carbonyl $T_1$ photolysis data, with
mean absolute deviations of ∼6 kJ/mol (Rowell et al., 2019). Comparisons of our B2GP-PLYP-D3/def2-TZVP thresholds with previous literature are provided in Tables S1–S6 of Supplementary Material and indicate excellent agreement with high-level correlated methods using accurate geometries. Note that, for $S_0$ energy differences between structurally similar molecules, as studied here, relative errors would be expected to be lower, approaching 4 kJ/mol 'chemical accuracy' (Pople, 1999).

Transition states were confirmed as first-order saddle points with a single imaginary frequency. Intrinsic reaction coordinate
(IRC) calculations were performed to ensure that the normal mode corresponding to the imaginary frequency connected the desired reactant and product minima. In order to reduce computational burden, IRC calculations were not performed where a TS had a structure and reaction coordinate directly analogous to a homologous molecule for which the reaction mechanism had been verified.

Schematic representation of the optimised saddle point structures are shown in the figures below. They are also represented
in Figs S2 – S7 and their Cartesian coordinates are provided as Supplementary Material. In general, only the lowest energy thresholds for each reaction class are discussed below, higher energy thresholds are reported as Supplementary Material.



## 3    Results and Discussion

The lowest energy calculated reaction thresholds for the up to nine $S_0$ reactions considered are reported in Table 1 for carbonyls in the dataset. Additional, higher energy, thresholds are reported as Supplementary Material. Asymptotic energies for NTI$a$ and NTI$b$, reproduced from Rowell et al. (2019) or, where available, the Active Thermochemical Tables (Ruscic et al., 2005; Ruscic, B. and Bross, D. H., 2020), are also included in Table 1.

Each reaction type is discussed separately below, with the three $H_2$–loss reactions discussed together. These data and available literature and experimental results are then analysed to determine chemically rationalised trends that will be applicable to larger carbonyls.

### 3.1    Decarbonylation (CO–loss)

Decarbonylation, the unimolecular loss of a CO molecule, is uncommon in gas-phase ketones. It has been observed in acetone, although the mechanism is unclear; Skorobogatov et al. (2002) postulated a direct mechanism, presumably via a TS, and roaming reactions have also been suggested in acetone (Goncharov et al., 2008; Saheb and Zokaie, 2018). Here, like previous studies (Saheb and Zokaie, 2018; So et al., 2018), we find no computational evidence for direct decarbonylation TSs in ketones.

In aldehydes, the formyl hydrogen is transferred to the main alkyl chain, forming CO and an alkane: R–(C=O)–H → CO + RH. Both roaming and TS decarbonylation mechanisms have been observed in gas-phase reactions of small aldehydes (Townsend et al., 2004; Houston and Kable, 2006; Heazlewood et al., 2008; Rubio-Lago et al., 2012; Yang et al., 2020). Figure 4 shows TS decarbonylation thresholds for the aldehydes in the dataset.

One trend apparent in Fig. 4 is that extension of the alkyl chain has little effect on decarbonylation threshold, with formaldehyde through pentanal all predicted to have ∼350 kJ/mol thresholds. The predicted decrease in threshold upon branching at the $\alpha$-position is <5 kJ/mol.

The decarbonylation thresholds in the $\alpha$,$\beta$-unsaturated enals, acrolein and methacrolein, are ∼10 kJ/mol higher than those for saturated aldehydes, as decarbonylation involves breaking a resonance stabilised C–C bond. Further delocalisation of the $\pi$ system increases the threshold to 375 kJ/mol for crotonaldehyde.

The $\alpha$-dicarbonyls are predicted to have the lowest decarbonylation thresholds, which is consistent with the two electron-withdrawing oxygen substituents reducing the electron density between the carbonyl moieties and weakening the $\alpha$-C–C bond. Four decarbonylation thresholds were found for glyoxal (Sect. S2 of Supplementary Material) with the lowest energy threshold of 225 kJ/mol corresponding to a non-planar TS and formation of formaldehyde and CO. This threshold is lower than the 251 kJ/mol threshold to form hydroxymethylene, HOCH, and CO. A similar non-planar TS is found for decarbonylation in methylglyoxal to form acetaldehyde, which has a slightly higher threshold of 235 kJ/mol. Here, substitution strengthens the central C–C bond due to electron-donation from the $CH_3$ substituent and there is also a steric penalty at the TS (Fig. S3a *vs.* S3e).





**Table 1.** Lowest energy zero-point vibrational energy corrected B2GP-PLYP-D3/def2-TZVP $S_0$ thresholds (kJ/mol) for the indicated unimolecular reactions of the carbonyls in the dataset (see text).

| | CO–loss | TF | NTIII | H$_2$–loss (*H positions*) (formyl+$\alpha$) | ($\alpha$+$\beta$) | ($\beta$+$\gamma$) | AE | Tautomerisation Keto–enol | Enal–ketene | Asymptotic[a] NTI$a$ | NTI$b$ |
|---|---|---|---|---|---|---|---|---|---|---|---|
| *Aldehydes:* | | | | | | | | | | | |
| Formaldehyde | 352 | – | – | – | – | – | – | – | – | 362.8±0.1 | 362.8±0.1 |
| Acetaldehyde | 352 | – | – | 337 | – | – | – | 281 | – | 346.4±0.3 | 367.8±0.4 |
| Propanal | 348 | 295 | 332 | 314 | 426 | – | – | 278 | – | 343.8±0.4 | 356 |
| Butanal | 351 | 299 | 311 | 320 | 410 | 516 | – | 282 | – | 341 | 360 |
| Pentanal | 350 | 294 | 310 | 317 | 409 | 499 | – | 281 | – | 345 | 362 |
| 2-Methylpropanal | 345 | 304 | 330 | 310 | 422 | – | – | 296 | – | 332 | 358 |
| 2,2-Dimethylpropanal | 342 | 306 | 322 | – | – | – | – | – | – | 329 | 358 |
| *Ketones:* | | | | | | | | | | | |
| Acetone | – | – | – | – | – | – | 363 | 275 | – | 346.6±0.5 | 346.6±0.5 |
| Butanone | – | – | 354 | – | 447 | – | 351 | 274 | – | 344.7±1.0 | 346 |
| Pentan-2-one | – | – | 316 | – | 416 | 513 | 347 | 270 | – | 338 | 342 |
| Pentan-3-one | – | – | 350 | – | 442 | – | 347 | 276 | – | 342 | 342 |
| *$\alpha,\beta$-Unsaturated:* | | | | | | | | | | | |
| Acrolein | 360 | 343 | 394 | 373 | – | – | – | 291 | 293 | 397.9±1.2 | 352 |
| Crotonaldehyde | 375 | 351 | 387 | 386 | – | – | – | 133[b] | 292 | 412 | 363 |
| Methacrolein | 362 | 338 | 379 | – | – | – | – | – | 285 | 381 | 363 |
| Methyl vinyl ketone | – | – | 417 | – | – | – | 329 | 270 | – | 393.4±1.0 | 338 |
| MIPK | – | – | 395 | – | – | – | 334 | – | – | 376 | 343 |
| *Dicarbonyls:* | | | | | | | | | | | |
| Glyoxal | 225[c] | 247[d] | – | – | – | – | – | – | – | 290.5±0.6 | 360.2±1.4 |
| Methylglyoxal | 235[e] | 330 | 400 | – | – | – | – | – | – | 293 | 354 |
| Diacetyl | – | – | 415 | – | – | – | – | – | – | 303.9±0.8 | 346 |
| *Carbohydrates:* | | | | | | | | | | | |
| Glycolaldehyde | 345 | 229[f] | – | 292 | 384[g] | – | – | 272 | – | 310 | 358 |

*a*: NTI$a$ and NTI$b$ are cleavage to the larger and smaller fragments. Values with uncertainties from the Active Thermochemical Tables (v. 1.22p) (Ruscic, B. and Bross, D. H., 2020; Ruscic et al., 2005), other values reproduced from Rowell et al. (2019); *b*: for a [1,5]-H atom shift to form but-1,3-dien-1-ol; *c*: for formation of formaldehyde + CO; *d*: for a 4-centre TS involving the two formyl H atoms; *e*: for formation of acetaldehyde + CO; *e*: The $\beta$-H for the TF transition state is from the OH moiety; *f*: The $\beta$-H for the TF transition state is from the OH moiety; *g*: The H$_2$–loss TS involves the OH hydrogen.

All calculated decarbonylation thresholds are accessible at the maximum tropospheric photon energy of 400 kJ/mol although, with the exception of formaldehyde (Moortgat et al., 1983), CO–loss QYs are low. For example, they are measured to be ~0.02
in acetaldehyde (Blacet et al., 1942) and ~0.01 for butanal (Blacet and Calvert, 1951).





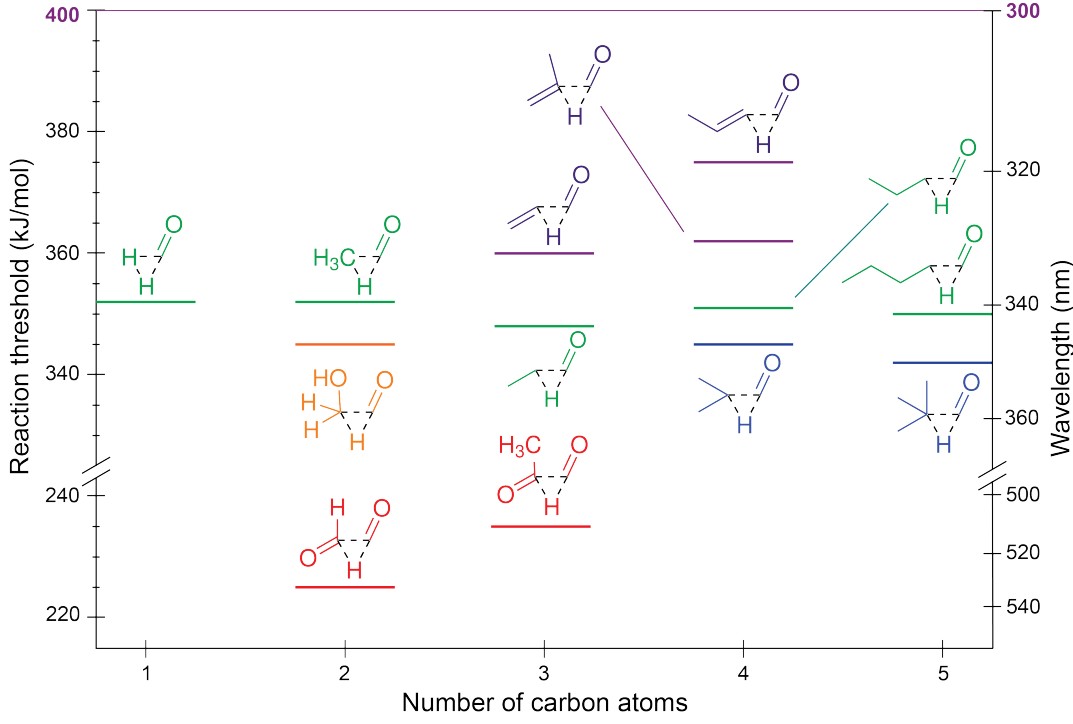

**Figure 4.** Zero-point vibrational energy corrected B2GP-PLYP-D3/def2-TZVP $S_0$ decarbonylation thresholds.

The thresholds for roaming decarbonylation reactions are linked to the $S_0$ NTI asymptotic energies (Andrews et al., 2013; Mauguière et al., 2015; Bowman and Houston, 2017). For example, the roaming threshold in formaldehdye is ~2 kJ/mol below the radical dissociation threshold (Townsend et al., 2004; Quinn et al., 2017), and both H and $CH_3$ roaming has been observed in acetaldehyde (Lee et al., 2014). As such, the majority of aldehydes in the dataset have roaming thresholds that
are similar to or lower in energy than the TS decarbonylation threshold, with formaldehyde a notable exception. Roaming pathways associated with NTI may also be present in ketones (Goncharov et al., 2008; Saheb and Zokaie, 2018). This suggests decarbonylation will be energetically accessible under tropospheric conditions for all carbonyls.

### 3.2 Triple fragmentation (TF)

We define the TF reaction as a concerted $S_0$ reaction forming three products via a single TS, where one of these products is $H_2$.
For example, the TF reaction in saturated aldehydes is: $H–CH_2–(C=O)H \rightarrow R=CH_2 + H_2 + CO$. For enals, the hydrocarbon product will be an alkyne. In dicarbonyls, TF forms $H_2$, CO, and either a second CO or ketene. Predicted $S_0$ TF thresholds are shown in Fig. 5. All of the TF reactions shown, except that for glyoxal, involve a 5-centre TS and H-loss from a $\beta$-hydrogen. In glyoxal, the 4-centre TS involves the two formyl hydrogens. The TSs are shown explicitly in Fig. S4 of the Supplementary Material, and are 'late', resembling the photolysis products. As a result, TF threshold energies are strongly influenced by the
stability of the products.



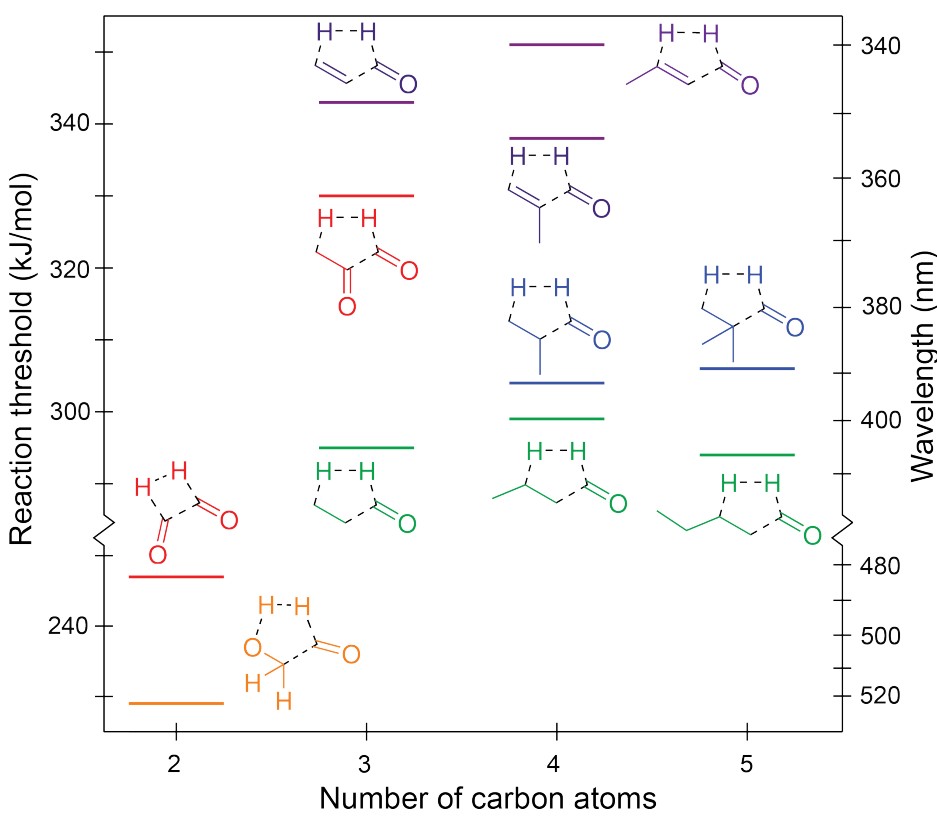

**Figure 5.** Zero-point vibrational energy corrected B2GP-PLYP-D3/def2-TZVP $S_0$ concerted triple fragmentation thresholds.

The TF thresholds for the saturated aldehydes are not significantly affected by chain extension, with thresholds of: 295, 299, and 294 kJ/mol for propanal, butanal, and pentanal, respectively. The effect of alkyl branching at the $\alpha$-position is inconsistent and small. It raises the thresholds to 304 kJ/mol for 2-methylpropanal and 306 kJ/mol for 2,2-dimethylpropanal, but lowers it for methacrolein (338 kJ/mol) compared to acrolein (343 kJ/mol). These differences are close to the likely accuracy of the B2GP-PLYP-D3 calculations.


The TF thresholds of the enals are significantly higher than those of the saturated aldehydes, at $\sim$340–350 kJ/mol. This increase is larger than the $\sim$10 kJ/mol increase seen in decarbonylation thresholds for breaking a delocalised C–C bond and arises because of bond angle strain in the 5-centre TF TSs of the $\alpha,\beta$ unsaturated species. As shown in Fig. S4, the C–C–C backbone angles in the TSs deviate significantly from their equilibrium values and the energetic penalty is increased, relative to saturated species, because of the rigidity of the delocalised C–C bonds.


The predicted $S_0$ TF thresholds of the two $\alpha$-dicarbonyls, glyoxal and methylglyoxal, differ markedly, by 83 kJ/mol. This arises because their TS structures and reaction products are qualitatively different (Figs S4j and S4k). Methylglyoxal has a 5-centre TS that involves strain across the O=C–C=O backbone and also forms a relatively high energy ketene product. Glyoxal,





in contrast, forms exothermically favourable products: $H_2$ and two CO molecules, resulting in a significantly lower $S_0$ TF
threshold.

Glycolaldehyde is an atypical carbonyl and has the lowest energy TF pathway, calculated at 229 kJ/mol. Here, the $H_2$–loss
channel arises from combination of the formyl-hydrogen and the OH-hydrogen, which is more labile than the backbone C–H
hydrogens. The C–OH angle in the glycolaldehdye TS also involves minimal ring strain (Fig. S4l). This pathway has not been
previously proposed for glycolaldehyde (Wallington et al., 2018; Bacher et al., 2001; Zhu and Zhu, 2010; Cui and Fang, 2011;
So et al., 2015) and occurs via a 5-centre TS, which is lower in energy than the 4-centre TF TS for $H_2$ formation from the
formyl and a $\beta$ $CH_2$ hydrogen. Indeed it is the lowest energy dissociation pathway we have found in gylcolaldehyde.

### 3.3  Norrish Type III $\beta$-H transfer reaction (NTIII)

Alongside the Norrish Type I and II reactions, the lesser known 'Norrish Type III' reaction (NTIII) was proposed by Zahra
and Noyes (1965) to explain the observation of acetaldehyde and propene as products of 3-methylbutan-2-one photoexcitation.
NTIII involves a 4-centre TS, with $\beta$-hydrogen transfer from the backbone to the carbonyl moiety leading to formation of an
aldehyde and an alkene, for example, $RCH_2CH_2(C=O)R' \rightarrow$ R–CH=$CH_2$ + H(C=O)R'. Experimentally, NTIII has been found
to be a minor channel (Zahra and Noyes, 1965). However, since the alkene product from NTIII can also be formed in other $S_0$
reactions (TF, $H_2$-loss), NTIII needs to be well understood to disambiguate these mechanisms.

Figure 6 shows our calculated NTIII thresholds. These vary across a large energy range, from 310–417 kJ/mol. In glyoxal,
methylglyoxal, and methyl vinyl ketone the NTIII thresholds are at or above the actinic maximum energy of 400 kJ/mol. The
NTIII thresholds of the other $\alpha,\beta$-unsaturated carbonyls lie between 380 and 400 kJ/mol. The high thresholds arise from reso-
nance stabilisation of the breaking C–C bond and suggest NTIII is unlikely to be important in these species under tropospheric
conditions.

The saturated carbonyls in Fig. 6 have NTIII thresholds in the range 310–355 kJ/mol. The NTIII thresholds decrease signif-
icantly when the main alkyl chain is lengthened past the $\beta$-position: there is a 38 kJ/mol lowering of the NTIII threshold from
butanone to pentan-2-one, and a 21 kJ/mol lowering from propanal to butanal. The effect of alkyl chain lengthening beyond the
$\beta$-position is also present in $\alpha,\beta$-unsaturated carbonyls, although it is smaller, with only a 7 kJ/mol decrease in threshold from
acrolein to crotonaldehyde. The NTIII threshold, however, is unchanged with further alkyl chain lengthening. For example,
butanal and pentanal have almost the same NTIII threshold (311 and 310 kJ/mol respectively), and there is only a 4 kJ/mol
difference in threshold between butanone and pentan-3-one. This suggests chain lengthening past the $\gamma$-position will not result
in any change to the reaction threshold. The NTIII threshold also appears to be independent of the 'spectator' alkyl substituent
in the ketones.

Branching at the $\alpha$-position decreases the NTIII threshold, though to a lesser extent than addition of a $\gamma$-carbon. The de-
creases are: 2 kJ/mol from propanal to 2-methylpropanal, 7 kJ/mol from 2-methylpropanal to 2,2-dimethylpropanal, 15 kJ/mol
from acrolein to methacrolein, and 23 kJ/mol ketone to methyl isopropenyl ketone. All but the smallest decrease in threshold
are outside the likely accuracy of the B2GP-PLYP-D3 calculations. The trends in NTIII threshold with both alkyl chain length



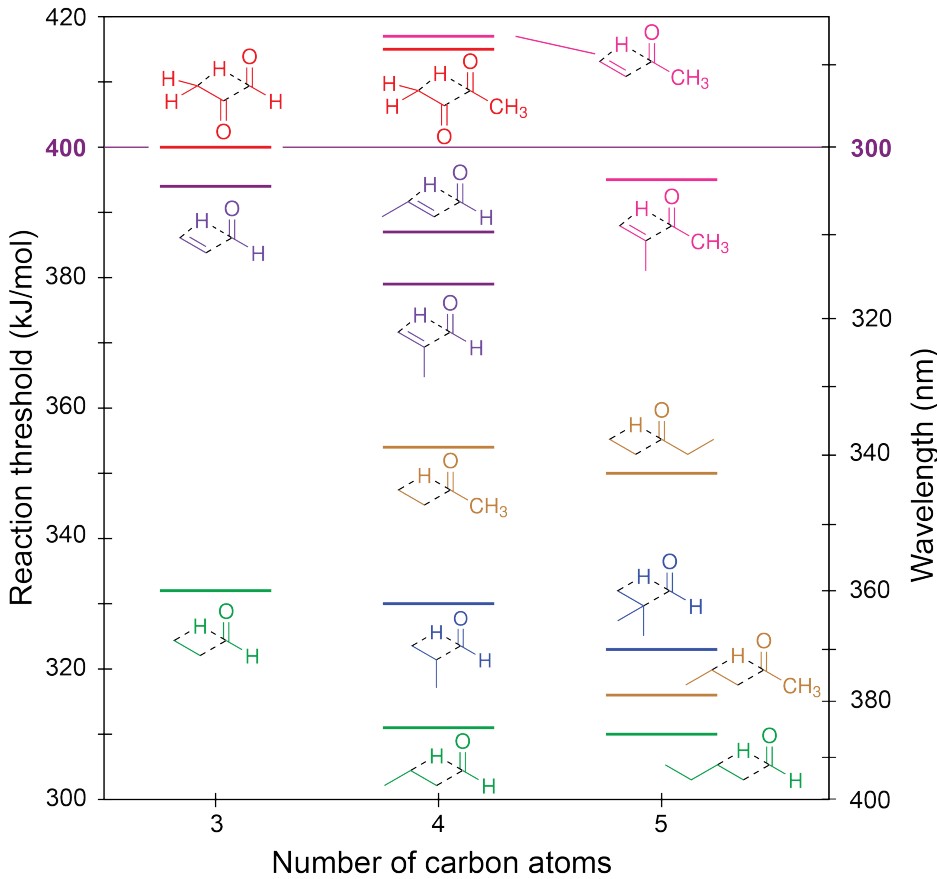

**Figure 6.** Zero-point vibrational energy corrected B2GP-PLYP-D3/def2-TZVP $S_0$ Norrish Type III thresholds.

and branching can be rationalised in terms of the alkene product formed: the relative stability of the alkene increases with increasing substitution about the double bond (Whangbo and Stewart, 1982).

### 3.4 Concerted 4-centre $H_2$–loss

$H_2$ can be formed on $S_0$ via a 4-centre TS where adjacent hydrogen atoms form an H–H bond and dissociate as $H_2$, leaving an unsaturated carbonyl product. In aldehydes these adjacent hydrogens can be the formyl and $\alpha$ hydrogens, but as the alkyl chain lengthens possibilities include hydrogens in the: $\alpha+\beta$, $\beta+\gamma$, etc., positions. These $H_2$–loss mechanisms can be distinguished by the point of unsaturation in the co-product.

    The $S_0$ reaction thresholds for the possible $H_2$–loss channels are given in Table 1 and are shown in Fig. 7, where solid

lines denote $H_2$–loss from the formyl and $\alpha$ positions; dashed lines, $H_2$–loss from the $\alpha$ and $\beta$ positions; and dot-dashed lines, $H_2$–loss from the $\beta$ and $\gamma$ positions. Note that the $y$-axis of Fig. 7 is broken into three energy sections to indicate energetic separation between the different mechanisms.



**Figure 7.** Zero-point vibrational energy corrected B2GP-PLYP-D3/def2-TZVP $S_0$ H$_2$–loss thresholds. *Solid lines*: formyl–H + $\alpha$–H; *Dashed lines*: $\alpha$–H + $\beta$–H; *Dot-dashed lines*: $\beta$–H + $\gamma$–H.





Figure 7 indicates that thresholds for $H_2$–loss from the $\alpha$ and $\beta$ positions are higher than from the formyl and $\alpha$ positions, with $H_2$–loss thresholds from the $\beta$ and $\gamma$ positions higher still. For a given carbonyl, the highest $H_2$–loss threshold is predicted

for removal of a hydrogen from the terminal carbon. Like the NTIII reaction, the TSs for 4-centre $H_2$–loss are 'late', and the threshold energies are related to the stability of the forming alkene product. Increasing substitution around the double-bond increases alkene stability (Whangbo and Stewart, 1982), so products with a terminal C=C bond are comparatively less stable than products from $H_2$–loss at other sites. For example, there is an approximately 20 kJ/mol decrease in threshold for formyl + $\alpha$ $H_2$–loss from acetaldehyde to propanal. Like NTIII, there is little effect on the formyl and $\alpha$ $H_2$–loss threshold upon further

chain extension. For example, the formyl and $\alpha$ $H_2$–loss thresholds for propanal, butanal and pentanal are all ∼315 kJ/mol. Branching at the $\alpha$ position also has little effect on the formyl and $\alpha$ $H_2$–loss threshold (*cf.* propanal and 2-methylpropanal). Similar trends are seen for $H_2$–loss from the $\alpha$ and $\beta$ and $\beta$ and $\gamma$ positions.

The results in Table 1 and Fig. 7 reinforce, for multiple carbonyl species, that only $H_2$–loss from the formyl and $\alpha$ positions is energetically accessible in the actinic energy range. Indeed, the 4-centre $H_2$–loss channel in acetaldehyde has recently been

observed experimentally under tropospherically relevant conditions (Harrison et al., 2019). In the absence of a formyl hydrogen, none of the ketone $H_2$–loss channels are accessible in the troposphere. Similarly, $H_2$–loss is not accessible in species lacking an $\alpha$-hydrogen, like 2,2-dimethylbutanal and methacrolein.

The thresholds for $H_2$–loss from the formyl and $\alpha$ positions in acrolein and crotonaldehyde are also in the actinic range, at 373 and 386 kJ/mol, respectively. These thresholds are significantly higher than those for the saturated aldehydes because

of the high energy of the product propadienone and 1,2-butadienone species and are close to the maximum actinic energy, suggesting $S_0$ $H_2$–loss is unlikely to be important in $\alpha,\beta$-unsaturated aldehydes.

Glycolaldehyde is calculated to have the lowest $H_2$–loss threshold, 292 kJ/mol, for loss of the formyl and $\alpha$ hydrogens. This can be rationalised in terms of the electron withdrawing nature of the OH stabilising the 4-centre TS and the hydroxyketene product. Glyolaldehdye is also the only carbonyl with an energetically accessible $\beta$-H loss channel. Loss of the OH and $\beta$

hydrogen forms glyoxal and $H_2$ with a 384 kJ/mol threshold, close to the actinic maximum energy.

### 3.5   Alkane/alkene elimination (AE)

In ketones, migration of an $\alpha$-H atom to the 'other' $\alpha$ carbon via a 4-centre TS can form an alkane and a ketene. For example, acetone can dissociate to methane and ketene. In methyl vinyl ketone, the unsaturation leads to alkene elimination and formation of ethene and ketene.

The $S_0$ reaction thresholds for the lowest energy AE channels are given in Table 1 and are shown in Fig. 8. In asymmetric ketones two AE channels are possible and these are described in Sect. S6 of the Supplementary Material.

Figure 8 shows our calculated AE thresholds vary from 329 to 363 kJ/mol, with all accessible under tropospheric conditions. The TSs for AE are shown in Fig. S6.

In linear unsaturated ketones, the AE TSs are 'late', with C–C breaking bond lengths over 1.8 Å. Naïvely, we would therefore

infer AE thresholds for these species will reflect the relative stability of the forming products. From tabulated 0 K enthalpies of formation (Ruscic et al., 2005; Ruscic, B. and Bross, D. H., 2020), although methyl ketene is 9 kJ/mol more stable than





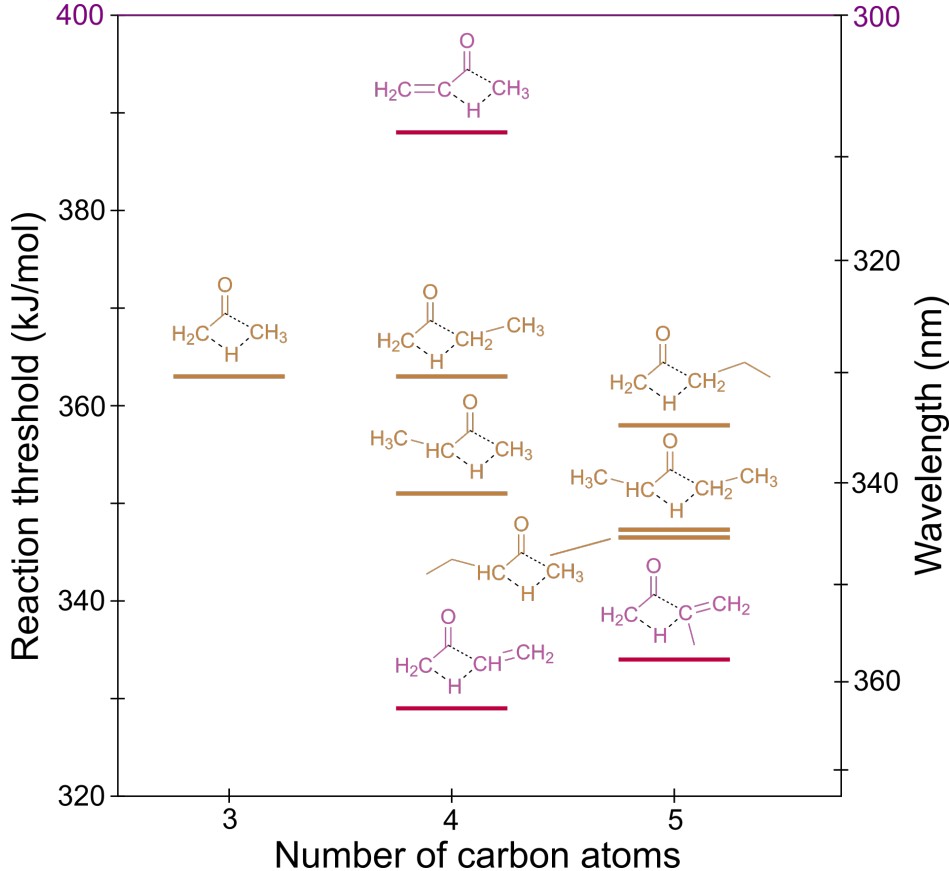

**Figure 8.** Zero-point vibrational energy corrected B2GP-PLYP-D3/def2-TZVP $S_0$ alkane/alkene elimination thresholds.

ketene, butanone is 17 kJ/mol more stable than acetone. This would imply a lower AE threshold in acetone. The AE threshold in butanone, however, is calculated to be 12 kJ/mol lower than that for acetone. The TS for AE in butanone (Fig. S6b) is marginally earlier than in acetone (Fig. S6a), suggesting electronic effects are responsible for the lower threshold. Only a 4

kJ/mol decrease in threshold, to 347 kJ/mol, is seen for production of ethylketene in pentan-2-one, indicating further chain lengthening has little effect. This threshold is the same as that calculated for pentan-3-one forming ethane and methylketene, that is, there is a small reduction in threshold energy on formation of a larger alkane. This is also seen in the alternate AE pathways (Sect. S6). As shown in Fig. S6, in linear unsaturated ketones, thresholds for producing ketene (via the alternate AE channels) are ~360 kJ/mol and thresholds for producing methylketene are ~350 kJ/mol. We expect these thresholds to be

generalisable to larger linear unsaturated ketones.

The lowest AE thresholds are for methyl vinyl ketone and methyl isopropenyl ketone (MIPK), both of which yield ketene. The AE TSs in these molecules are much 'tighter' than in the linear unsaturated ketones (Fig. S6). Here there is resonance stabilisation of the saddle points, leading to thresholds of 329 and 334 kJ/mol, respectively.



### 3.6 Keto–enol tautomerisation

Carbonyls can exist in two tautomeric forms: a keto form (encompassing, here, both ketones and aldehydes), and an enol form where an H atom has transferred to the carbonyl oxygen, forming an OH substituent and a point of unsaturation. Keto–enol tautomerisation is known to occur as a dynamic equilibrium in $S_0$ carbonyls in aqueous solution at room temperature, although the keto tautomer is thermodynamically favoured (Keeffe et al., 1988). Keto–enol tautomerisation has been observed in gas phase photolysis experiments on acetaldehyde (Clubb et al., 2012; Shaw et al., 2018) and the authors suggest it may occur in

many other carbonyls.

The lowest energy calculated keto–enol tautomerisation thresholds for the relevant carbonyls in the dataset are given in Table 1 and shown in Fig. 9. A keto–enol TS involving an $\alpha$ hydrogen, that is, a [1,2]-H atom shift, was found in all of these species. In crotonaldehyde, however, the lowest energy threshold, 133 kJ/mol, was for a [1,5]-H atom shift involving a $\gamma$ hydrogen and this is less than half the 294 kJ/mol threshold we predict for a [1,2]-H atom shift. This mechanism is analogous to the [1,5]-H

atom shift in the Norrish Type II reaction. Here, however, because it occurs in an enal, conjugation prevents the bond between the $\alpha$- and $\beta$-carbons from breaking and, instead, but-1,3-diene-1-ol, is formed.

There may be multiple possible [1,2]-H atom shift keto–enol tautomerisation pathways for a given carbonyl. These correspond to formation of geometric isomers (e.g. *cis*- or *trans*-enols) or, in asymmetric ketones, tautomerisation involving hydrogens from either alkyl substituent. These additional keto–enol tautomerisation thresholds are reported in Table S7 and

all optimised saddle point geometries are shown in Fig. S7 of the Supplementary Material. As described in more detail in Sect. S7, due to steric factors, thresholds for tautomerisation to *trans*-enols are typically ∼20 kJ/mol lower than those to the corresponding *cis*-enol.

As shown in Fig. 9, the keto–enol tautomerisation thresholds for linear aldehydes lie in a narrow energy range (278–281 kJ/mol), indicating chain extension has no effect on threshold as long as the bulky alkyl group can be oriented *trans* to the

enol OH group. This is not the case for the $\alpha$-branched 2-methylpropanal, and the steric penalty leads to the highest keto–enol tautomerisation threshold calculated here (296 kJ/mol).

The tautomerisation thresholds for ketones are also in a narrow range (270–276 kJ/mol), ∼5 kJ/mol lower than the corresponding aldehydes. As shown in Table S7 this extends to the alternate pathway in asymmetrically substituted ketones, which have thresholds for formation of the alternate *trans*-enol within ∼1 kJ/mol of those shown in Fig. 9.

In $\alpha,\beta$-unsaturated carbonyls, the keto–enol tautomerisation thresholds are low when tautomerisation involves a H-atom from an aliphatic group, for example, the $CH_3$ group in methyl vinyl ketone and crotonaldehyde. Tautomerisation thresholds involving olefinc H-atoms, however, are significantly higher, for example 292 kJ/mol in acrolein, reflecting the relatively unstable propadienol product.

Glycolaldehyde has an –OH electron withdrawing functional group. For this molecule the tautomerisation threshold is

predicted to be lowered by 8 kJ/mol compared to acetaldehyde, that is, the electron withdrawing group stabilises the TS to the forming enol.

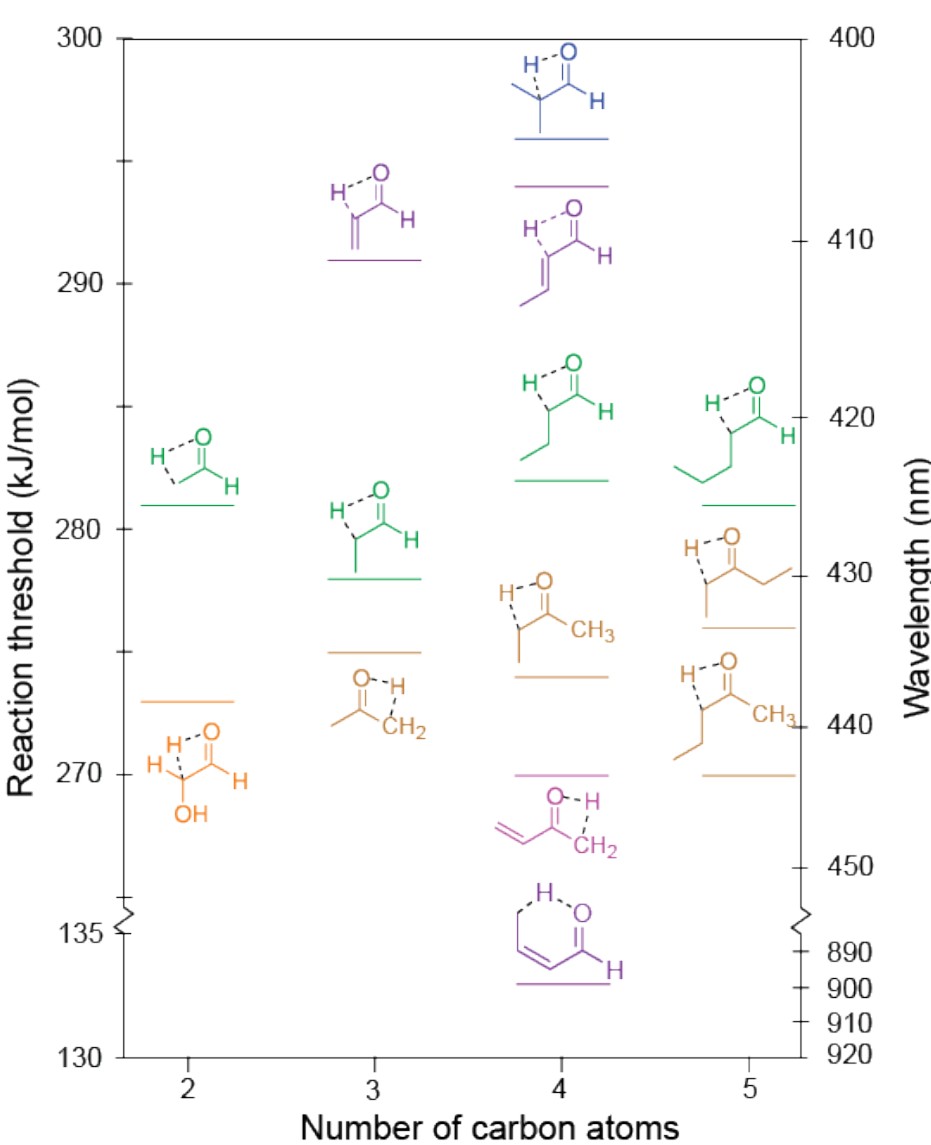

**Figure 9.** Zero-point vibrational energy corrected B2GP-PLYP-D3/def2-TZVP $S_0$ keto–enol tautomerisation thresholds.



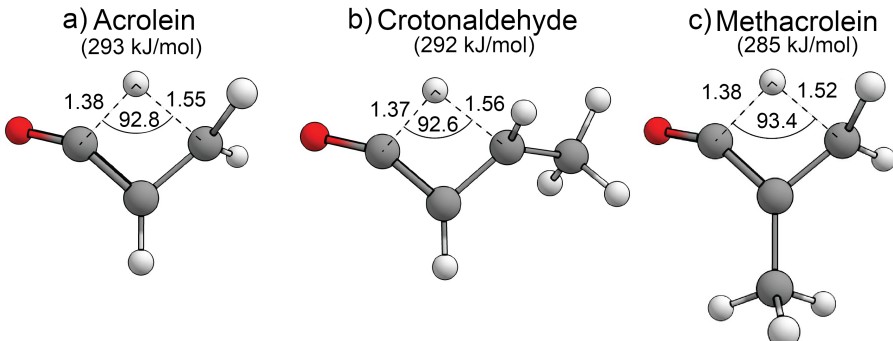

**Figure 10.** Optimised B2GP-PLYP-D3/def2-TZVP $S_0$ TSs and zero-point vibrational energy corrected threshold energies for enal–ketene tautomerisation, as shown. Key structural parameters in Å and degrees.

Notably, all calculated keto–enol tautomerisation thresholds are significantly below the maximum actinic photon energy. We expect this pathway to be energetically accessible in all carbonyls with an $\alpha$ hydrogen and appropriate unsaturated species with a $\gamma$ hydrogen. Moreover, all linear aldehydes and ketones are calculated to have keto–enol tautomerisation thresholds close to, or below, that of acetaldehyde. Given the experimental observation of keto–enol tautomerisation in acetaldehyde (Clubb et al., 2012; Shaw et al., 2018) and in acetone and methyl vinyl ketone (Couch et al., 2021), this tautomerisation may be important under tropospheric conditions in these species.

### 3.7 Enal–ketene tautomerisation

There has been recent interest in the formation of ketenes as atypical and relatively uncharacterised products of carbonyl photolysis (Harrison et al., 2019; Toulson et al., 2018). As seen above, the formyl + $\alpha$ $H_2$–loss mechanism forms ketenes in aldehydes and the AE mechanism forms ketenes in ketones. In enals there is also an $S_0$ tautomerisation mechanism involving a [1,3]-H shift of the formyl-hydrogen to the $\beta$-carbon that can form ketenes. First order saddle points have been optimised for enal–ketene tautomerisation in acrolein, crotonaldehyde and methacrolein. These are shown in Fig. 10, together with the calculated B2GP-PLYP-D3 threshold energies. The enal–ketene tautomerisation threshold in methacrolein, 285 kJ/mol, is ~13 kJ/mol lower in energy than the 298.7 kJ/mol G3X-K//M06-2X/6-31G(2df,p) threshold previously calculated by So et al. (2018). This difference is consistent with the variation between B2GP-PLYP-D3 and G3X-K thresholds for other reactions considered in methacrolein and methyl vinyl ketone (see Supplementary Material) and may be due to the treatment of dispersion in the saddle point geometries.

Figure 10 shows the key structural parameters describing the TSs for $S_0$ enal–ketene tautomerisation are almost identical for the three enals in the dataset. This indicates $\alpha$-branching and main alkyl chain extension have negligible effect on the TS geometry and little effect on the threshold energy. The predicted enal–ketene tautomerisation thresholds are all well below the maximum actinic energy of 400 kJ/mol and suggest that, in appropriate species, enal–ketene tautomerisation may be competitive in the troposphere.





## 4 Competition between carbonyl $S_0$ reactions

Our calculated $S_0$ thresholds can be compared to relevant experimental results, to validate the calculations, generalise our results and to predict the most important $S_0$ reactions for each class of carbonyl. Indeed, the seven reaction thresholds depicted for butanal in Fig. S1 are illustrative of the energetic relevance of these channels in other saturated aldehydes.

### 4.1 Saturated aldehydes

For saturated aldehydes, the lowest energy dissociation pathway on $S_0$ is concerted TF, with a threshold of ∼300 kJ/mol.
At wavelengths and internal energies where excited state chemistry is energetically inaccessible, TF may therefore be a significant photo-induced mechanism. Triple fragmentation has been observed as a primary photolysis mechanism in propanal and 2-methylpropanal, with QYs of 4% and 9%, respectively, at 1 atm pressure of $N_2$ (Kharazmi, 2019). The increased QY for 2-methylpropanal was attributed to increased reaction path degeneracy rather than a lower reaction threshold. This suggests TF QYs will be higher in branched aldehydes, in proportion to the ratio of $\beta$-hydrogens. For example, we predict 2,2-
dimethylpropanal, with nine $\beta$-hydrogens to have a higher TF QY than 2-methylpropanal, with six $\beta$-hydrogens, and propanal, with three $\beta$-hydrogens.

The next lowest energy $S_0$ dissociation pathway is either $H_2$-loss from the formyl and $\alpha$–H atoms in the smaller species (acetaldehyde, propanal, methylpropanal) or NTIII in the larger species (butanal, pentanal, 2,2-dimethylpropanal). For formyl + $\alpha$ $H_2$-loss, acetaldehyde has three $\alpha$ hydrogens and the highest threshold at 337 kJ/mol. This channel has been observed
experimentally, with a QY of ∼1% at 305 nm (392 kJ/mol) and 1 atm of $N_2$ (Harrison et al., 2019). The equivalent thresholds for the other saturated aldehydes, which have two rather than three $\alpha$ hydrogen atoms, are significantly lower, at ∼315 kJ/mol. Thus we expect $H_2$–loss reactions to be present in other saturated aldehydes with similar, or slightly higher QYs than in acetaldehyde. Indeed, under tropospheric conditions, methylketene and dimethylketene have been observed as minor products following photolysis of propanal and 2-methylpropanal, respectively (Kharazmi, 2019). In saturated aldehydes, the NTIII
thresholds are similar to those for $H_2$–loss. Experimental QYs for this pathway, however, have not been reported. The NTIII mechanism shares common products with TF (alkene) and decarbonylation (CO), which may complicate its experimental identification. Our calculations suggest NTIII is energetically competitive, and it may be important in interpreting photolysis QYs of saturated aldehydes.

With the exception of formaldehyde, the next lowest energy $S_0$ dissociation threshold is for ground state NTI$a$, that is,
$\alpha$-bond cleavage to the alkyl substituent. This reaction is typically assumed to occur only on $T_1$ but the $S_0$ reaction has been identified in formaldehyde (Quinn et al., 2017), acetaldehyde (Heazlewood et al., 2009; Amaral et al., 2010) and acetone (Lee et al., 2017). Because it is barrierless we expect $S_0$ NTI$a$ to dominate as internal energy increases above its threshold. Thus, we expect to observe NTI$a$ dissociation products at energies well below the $T_1$ threshold.

Decarbonylation (CO–loss) has the highest calculated $S_0$ dissociation thresholds for all the saturated carbonyls in the dataset,
bar formaldehyde, for which the decarbonylation threshold is slightly below that of NTI. All decarbonylation thresholds are predicted to be 340–350 kJ/mol and experimental CO–loss QYs are correspondingly low: in the actinic energy range, the



atmospheric pressure decarbonylation QY for acetaldehyde is $\sim$0.5% (Moortgat et al., 2010; Warneck and Moortgat, 2012) and $\sim$0.2% for 2-methylbutanal (Gruver and Calvert, 1958). For larger saturated aldehydes, we predict other $S_0$ pathways will be favoured over decarbonylation, which will be at most a minor channel.

Our calculations predict keto–enol tautomerisation to be the lowest energy pathway on $S_0$ for all saturated aldehydes, lying about 20 kJ/mol below TF. Unlike dissociation, however, tautomerisation is reversible. Collisional cooling into the enol well therefore competes with tautomerisation back to the parent aldehyde (Andrews et al., 2012). Dissociation of the aldehyde reduces its concentration and hence reduces the rate of enol formation. Photo-initiated keto–enol tautomerisation has only been observed in acetaldehyde (Clubb et al., 2012; Shaw et al., 2018), where TF is not available. Whilst it is likely keto–

enol tautomerisation is occurring in other saturated aldehydes, QYs are likely small and the reactivity of the enol will make characterisation difficult.

### 4.2    Saturated ketones

Following excitation by an actinic photon, the relevant $S_0$ chemistry of ketones is much simpler than that of aldehydes. No TS for direct decarbonylation was found and the absence of a formyl H–atom in ketones removes the TF and formyl + $\alpha$ H$_2$–loss

pathways. The only $S_0$ dissociation pathways in ketones are NTIII, AE and NTI. The thresholds for AE and NTI are similar in the ketones considered, at $\sim$340 kJ/mol. Given NTI is barrierless, it is likely to dominate over AE. We therefore expect NTI to be the dominant dissociation mechanism in acetone, butanone and pentan-2-one, where it has a lower threshold than NTIII, although NTIII may be important in pentan-2-one and in larger saturated ketones, where there is alkyl chain extension past the $\beta$-hydrogen involved in the reaction. For example, the production of acetaldehyde from 3-methylbutan-2-one is one of the few

experimental examples of the NTIII mechanism in the literature (Zahra and Noyes, 1965).

     The keto–enol tautomerisation thresholds for saturated ketones are slightly lower than in saturated aldehydes. This, combined with the absence of low energy $S_0$ dissociation pathways and the experimental observation of keto–enol tautomerisation in acetone (Couch et al., 2021), suggests photo-initiated tautomerisation may be important in ketones.

### 4.3    $\alpha,\beta$-Unsaturated carbonyls

In general, because of resonance stabilisation of the bond between $\alpha$- and formyl-carbons, we calculate higher $S_0$ thresholds in $\alpha,\beta$-unsaturated carbonyls than in equivalent saturated carbonyls. However, excited state NTI$a$ thresholds are also elevated and all excited state NTI thresholds are close to, or above, the maximum available actinic energy (Rowell et al., 2019). Thus any photolysis must occur on $S_0$. Further, unlike many other carbonyls, the electronic structure of the $\alpha,\beta$-unsaturated carbonyls promotes rapid electronic relaxation from $S_1$ to $S_0$ (Lee et al., 2007; Schalk et al., 2014; Cao and Xie, 2016), suggesting high

$S_0$ internal energies and therefore relatively high probability of $S_0$ reaction.

     Triple fragmentation is the lowest energy $S_0$ dissociation pathway for the unsaturated aldehydes in the dataset. The TF thresholds are $\sim$40 kJ/mol higher than in saturated aldehydes and are slightly lower than the $S_0$ NTI$b$ asymptotic energies and decarbonylation thresholds. The other possible $S_0$ dissociation reactions (H$_2$-loss, NTIII and $S_0$ NTI$a$) have thresholds >370 kJ/mol and are unlikely to be significant under tropospheric conditions.





Unsaturated ketones, like their saturated counterparts, have fewer $S_0$ dissociation pathways, with only NTIII, AE, NTI$a$ and NTI$b$ possible. In methyl vinyl ketone, AE has the lowest threshold, at 329 kJ/mol, 9 kJ/mol lower than the NTI$b$ threshold. NTI$b$ dissociation, however, is barrierless. The 334 kJ/mol threshold for AE is the lowest energy $S_0$ pathway in methyl iso-propenyl ketone, 10 kJ/mol lower than the threshold for NTI$b$ dissociation. Although NTI$b$ will dominate at higher energies, there may be a small energy window where AE may be important, although it has not been experimentally observed. Similarly,

there may be an energy window for AE in methyl vinyl ketone. The other possible dissociations in methyl vinyl ketone, NTI$a$ and NTIII, have thresholds >375 kJ/mol and are unlikely to be important under tropospheric conditions.

Given the lack of low energy ground or excited state dissociation pathways, we expect photolysis QYs for $\alpha,\beta$-unsaturated carbonyls to be small. Indeed the total photolysis QY of methacrolein is around 1% at atmospheric pressure (Raber and Moortgat, 1996). Photo-induced keto–enol and enal–ketene tautomerisations, however, have thresholds under 300 kJ/mol, with

crotonaldhyde having a lowest energy keto–enol tautomerisation threshold of 133 kJ/mol. Tautomerisation has been observed in methyl vinyl ketone (Couch et al., 2021), and it may therefore be competitive for appropriate $\alpha,\beta$-unsaturated species (So et al., 2018). The higher energy enol and ketene isomers may be collisionally stabilised under tropospheric conditions, although they are significantly more reactive and will be difficult to experimentally isolate.

### 4.4    Other carbonyls

The $\alpha$-dicarbonyls have low energy $\pi_+^*$ excited states (Arnett et al., 1974; Dykstra and Schaefer, 1976), red-shifted absorption spectra compared to other carbonyls (Fig. 1) and weakened $\alpha$-C–C bonds due to the electron withdrawing nature of the two C=O chromophores. As a result, the excited state NTI$a$ thresholds in dicarbonyls are lowered to $\sim$390 kJ/mol on $S_1$ and $\sim$300 kJ/mol on $T_1$ (Rowell et al., 2019), with both accessible in the troposphere. The $S_0$ asymptotic energies for NTI$a$ are slightly lower than the triplet thresholds, giving an energetic window for $S_0$ radical dissociation. Indeed, in glyoxal, two distinct,

wavelength-dependent mechanisms of HCO$^\bullet$ formation have been observed and attributed to dissociation on two electronic states (Chen and Zhu, 2003; Kao et al., 2004; Salter et al., 2013). Decarbonylation to form $H_2CO + CO$ and TF to form $H_2 + 2CO$ have the lowest energy thresholds in glyoxal, at 225 and 247 kJ/mol, respectively. Indeed, CO is known to form following irradiation of glyoxal at energies below 272 kJ/mol (Loge and Parmenter, 1981; Hepburn et al., 1983; Burak et al., 1987; Dobeck et al., 1999). Decarbonylation in methylglyoxal also has a low threshold energy, 235 kJ/mol, and we expect it to be

the dominant $S_0$ dissociation mechanism. In diacetyl, without a formyl hydrogen, NTI$a$ has the lowest dissociation threshold. We therefore predict, for larger $\alpha$-dicarbonyls, there will be an actinic window from $\sim$300–266 kJ/mol with no accessible dissociation pathways. In this case, collisional cooling and thermalisation to the parent carbonyl is the likely fate.

The remaining carbonyl in the dataset is glycolaldehyde, with an actinic range between 352 and 400 kJ/mol (Bacher et al., 2001). The NTI$a$ thresholds on $S_1$ and $T_1$ for glycolaldehyde are 379 and 334 kJ/mol, respectively (Rowell et al., 2019), and

this channel dominates at all actinic energies (Bacher et al., 2001; Zhu and Zhu, 2010), with cleavage of the C–OH bond to form $^\bullet$OH also reported (Zhu and Zhu, 2010). Thus, despite having the lowest calculated TF and $H_2$–loss thresholds, these reactions have not been experimentally observed in glycolaldehyde. Keto–enol tautomerisation to 1,2-ethenediol, for which





we calculate a threshold of 272 kJ/mol, has been postulated as an atmospheric route to formation of $HO_2^\bullet$ and formic acid (So et al., 2015), although it has not been experimentally observed.

## 5 Tropospheric relevance of $S_0$ reactions

A few guiding principles can be used to determine which of the possible $S_0$ pathways may be tropospherically relevant in a given carbonyl following absorption of an actinic photon.

- Reactions with thresholds greater than the actinic maximum energy of 400 kJ/mol are inaccessible at tropospheric photon energies. This rules out most $S_1$ reactions, with exceptions of NTI$a$ dissociation in glycolaldehyde and NTI$b$ in methyl vinyl ketone and methyl isopropenyl ketone (Haas, 2004; Rowell et al., 2019). $T_1$ NTI dissociations *are* accessible. $T_1$ NTI$b$ thresholds are close to the maximum available energy of 400 kJ/mol and have negligible contribution to actinic photolysis (Zhu et al., 2009), $T_1$ NTI$a$ thresholds are generally lower (Rowell et al., 2019), and this reaction dominates the photolysis QY of small carbonyls (Kirkbride and Norrish, 1931; Zhu et al., 2009). In larger carbonyls (alkyl chains lengths $\geq 4$), excited state NTII intramolecular $\gamma$-H abstraction is also accessible (Wagner and Zepp, 1972; Wagner and Klán, 2004; Zhu et al., 2009).

- For photon energies above their thresholds, $T_1$ reactions are fast and dominate photolysis (Kirkbride and Norrish, 1931; Zhu et al., 2009). If the photon energy is near threshold, non-radiative transitions and $S_0$ reaction may be competitive with excited state reaction. For photon energies (or collisional cooling on $S_1$ or $T_1$) below any excited state threshold, any photolysis must occur on $S_0$. There is significant overlap of the absorption spectrum of most carbonyls with these lower energy photons (Fig. 1) and $S_0$ reactions have been observed in saturated carbonyls under these conditions (Heazlewood et al., 2009; Amaral et al., 2010; Andrews et al., 2012; Tsai et al., 2015; Quinn et al., 2017; Toulson et al., 2018).

- For an $S_0$ process to be tropospherically important, its rate must be competitive with collisional cooling and thermal equilibrium. Only $S_0$ dissociations with thresholds <350 kJ/mol have been experimentally observed following photoexcitation of aldehydes in 1 atm of $N_2$ (Clubb et al., 2012; Shaw et al., 2018; Harrison et al., 2019). This suggests the most important $S_0$ dissociations are those with the lowest thresholds, that is, TF if available, NTI in ketones and selected AE, NTIII and formyl+$\alpha$ $H_2$–loss reactions.

- Although $S_0$ keto–enol and enal–ketene tautomerisations have the lowest calculated thresholds, these are reversible reactions where collisional stabilisation of the tautomer competes with isomerisation back to and dissociation of the parent carbonyl. Tautomerisation is therefore unlikely to be important if there are low energy $S_0$ dissociation pathways. It will be important in the absence of such a channel, for example, in acetaldehyde (Clubb et al., 2012; Shaw et al., 2018), some ketones and some $\alpha,\beta$-unsaturated carbonyls.

The atmospheric importance of photo-initiated $S_0$ reactions is as yet unknown, however, $S_0$ reactions are likely to be broadly accessible under tropospheric conditions for all carbonyls. They are therefore likely in regions of the troposphere with high



carbonyl concentrations: in highly polluted environments, where carbonyls are both directly emitted and are oxidation products
of other VOCs, and in unpolluted, forested environments, where oxidation of BVOCs leads to unsaturated carbonyls like
methacrolein and methyl vinyl ketone (Kesselmeier and Staudt, 1999; Millet et al., 2010; Chen et al., 2014). Carbonyls are also
important species in the marine boundary layer, although there are significant discrepancies between observed and modelled
concentrations (Vigouroux et al., 2009; Anderson et al., 2017; Ahn et al., 2019).

Photo-initiated $S_0$ reactions of carbonyls have a number of atmospheric consequences. The majority of reactions lead to
molecular, rather than radical, products and this may effect modelled radical quantum yields and therefore radical propagation
reactions. Many of the $S_0$ reaction products are also 'unexpected', and form unsaturated species such as alkenes, alkynes, enols
and ketenes that are more reactive than their parent carbonyl, particularly to addition reactions, for example, with $^\bullet$OH and
NOx radicals and even atmospheric $H_2O$ (Atkinson et al., 2006; So et al., 2014; Kahan et al., 2013). We suggest ground state
reactions should be considered whenever unexpected products are found in the laboratory photolysis of carbonyls and when
products are observed following photolysis below excited state thresholds.

The experimental observations of photo-initiated $S_0$ reactions in carbonyls described in this paper indicate the parent car-
bonyl is returning to the ground electronic state with energies close to the original photon energy. In addition to dissociation
and isomerisation, these vibrationally 'hot' $S_0$ molecules could also react with other atmospheric species: their high internal
energy makes otherwise inaccessible reactions energetically feasible. For example, internally excited carbonyls could undergo
rapid bimolecular reaction with species such as $O_2$, $H_2O$, $^\bullet$OH, NOx and $NH_3$. The possibility of a reaction between $O_2$ and
internally hot acetaldehyde, formed following excitation at 248 nm, considerably above the actinic range, was speculated by
Morajkar et al. (2014), although this was not further investigated or resolved. Such bimolecular reactions may lead to radical
species and we suggest they may be responsible for radical QYs observed following photolysis at energies below the NTI
thresholds in formaldehyde (Horowitz and Calvert, 1978; Moortgat and Warneck, 1979; Valachovic et al., 2000). In this sense,
whilst the *total* experimental radical QY may be correct, the reaction mechanism is not simply unimolecular dissociation and
radical QYs from bimolecular reaction will have different pressure dependence to those from unimolecular dissociation.

The molecular products of some of the $S_0$ reactions considered are also formed with extremely high internal energy. Exper-
imentally, the roaming pathway in formaldehdye forms $H_2$ with up to 12 quanta of vibrational excitation (Quinn et al., 2017).
Similarly, $CH_4$ from roaming in acetaldehyde is formed with internal energies up to 95% of the CH bond dissociation energy
(Heazlewood et al., 2008). Although its mechanism has not been elucidated, the formyl+$\alpha$ $H_2$-loss channel in acetaldehyde
has been shown experimentally to yield internally hot ketene, with energy $> 150$ kJ/mol (Harrison et al., 2019). These 'hot'
molecular products could also potentially undergo bimolecular reactions in the atmosphere and contribute toward radical QYs.

There are two cases where photo-initiated $S_0$ reactions of carbonyls have immediate tropospheric implications and missing
or underestimated photochemical sources have previously been speculated. These are the formation of organic acids and the
photolytic generation of molecular hydrogen, $H_2$. In both polluted and pristine regions there are discrepancies between the
predictions of atmospheric models and field measurements of the concentrations of organic acids (Cady-Pereira et al., 2014;
Yuan et al., 2015; Millet et al., 2015). Depending on the type of model and the estimation of soil uptake, there are also large
differences in estimates of global photochemical production of $H_2$ (Novelli, 1999; Rhee et al., 2006; Price et al., 2007; Xiao





et al., 2007; Ehhalt and Rohrer, 2009; Yashiro et al., 2011; Patterson et al., 2020). Notably, in 'top-down' models, Rhee et al.
(2006) and Xiao et al. (2007) propose increased $H_2$ production from photolysis of oxidized non-methane VOCs.

The tautomerisation of acetaldehyde to vinyl alcohol, addition of atmospheric $^\bullet$OH and subsequent oxidation has been modelled and shown to produce significant global tropospheric formic acid and was found to be the dominant mechanism for formic acid formation in the marine boundary layer (Shaw et al., 2018). This one reaction, however, is not sufficient to explain the factor of two discrepancy between modelled and measured global organic acid concentrations (Shaw et al., 2018). Keto–
enol isomerisation is present in almost all carbonyls we have considered, having amongst the lowest $S_0$ reaction thresholds. Similar reactions involving other unsaturated products, for example alkenes and ketenes (Atkinson et al., 2006; Kahan et al., 2013), will also lead to organic acid formation. We expect these reactions to be energetically accessible in most atmospheric carbonyls and their cumulative effect may address the modelled deficit (Shaw et al., 2018; So et al., 2018). Understanding the mechanism of formation of organic acids in the troposphere has further application to secondary aerosol formation; the
higher oxygen content of organic acids reduces their volatility and, as proton donors, they are believed to be key species in the nucleation and growth of atmospheric particles (Zhang et al., 2004; Bianchi et al., 2016; Tröstl et al., 2016; Liu et al., 2021).

Molecular hydrogen is an important atmospheric reducing agent and is an indirect greenhouse gas because it increases the atmospheric lifetime of $CH_4$ (Ehhalt and Rohrer, 2009). Current understanding indicates the major photochemical source of $H_2$ in the atmosphere is photolysis of formaldehyde, which accounts for at least half of the photochemically generated $H_2$
(Hauglustaine, 2002). The mechanisms generating the other half are unknown (Grant et al., 2010). The energetically accessible $S_0$ TF and/or $H_2$-loss reactions present in all aldehydes in our dataset, and expected in all atmospheric aldehydes, provide primary photolysis routes to $H_2$ that have not previously been considered. By better understanding the current atmospheric $H_2$ budget, we are better placed to model any future increase in atmospheric $H_2$, for example, due to leakage of $H_2$ in any transition to a hydrogen economy.

**5.1 Challenges to address**

To definitively answer the question 'how important is the photo-initiated ground state chemistry of carbonyls in the atmosphere?' a number of challenges need to be addressed. Experimentally these include observation of primary products formed on $S_0$ and characterisation of $S_0$ QYs. An understanding of the competition between collisional cooling and reaction under tropospheric conditions is also required.

Primary photolysis products can be observed in single molecule experiments, for example, Harrison et al. (2019) recently observed the formyl+$\alpha$ $H_2$-loss channel in acetaldehyde using velocity map ion-imaging. These experiments, however, do not provide absolute QYs. An additional challenge is that many of the products formed are either transient and difficult to observe under atmospheric conditions or can be formed from multiple reaction pathways. Determination of $S_0$ QYs therefore relies on master equation modelling and hence on an accurate characterisation of reaction mechanisms. The results in this paper are
a necessary step toward such modelling and will inform the interpretation of photolysis experiments on individual carbonyls, box models of specific locations and ultimately global chemical transport models.



Very little is known about collisional cooling of highly excited ground state molecules. Modelling of experimental photolysis QYs of acetaldehyde, using a simple exponential-down model, suggests an average loss of 150 cm$^{-1}$ ($\sim$1.8 kJ/mol) internal energy per collision with $N_2$ (Andrews et al., 2012). The accuracy of this model and the generality of this result to other carbonyls and their isomers are yet to be tested. Nevertheless, it suggests of the order of 200 collisions are required to thermalise carbonyls after absorption of an actinic photon. Given collision frequencies of the order of $10^9$ s$^{-1}$ at atmospheric pressure, unimolecular reactions with rate coefficients of magnitude $10^7$ or more are likely to be competitive under tropospheric conditions. For example, for deuterated acetaldehyde following excitation by 322.9 nm light, Heazlewood et al. (2011) calculated unimolecular rate coefficients for $S_0$ NTI dissociation, decarbonylation, formyl+$\alpha$ $H_2$-loss and keto–enol tautomerisation of approximately $1 \times 10^7$, $1 \times 10^6$, $2 \times 10^6$, and $2 \times 10^7$ s$^{-1}$, respectively. There is experimental evidence for all of these $S_0$ reactions under tropospheric conditions (Heazlewood et al., 2009; Horowitz and Calvert, 1982; Heazlewood et al., 2008; Moortgat et al., 2010; Harrison et al., 2019; Clubb et al., 2012; Shaw et al., 2018), although decarbonylation and formyl+$\alpha$ $H_2$-loss are minor channels. Many of the $S_0$ thresholds in Table 1 are lower than those for acetaldehyde, implying higher reaction rate coefficients, although these are yet to be calculated,

Meeting these challenges will enable the tropospheric importance of photo-initiated $S_0$ reactions in individual carbonyls to be determined. Even if these individual reactions have small QY, their presence in all atmospheric carbonyls may lead to cumulative products that may be atmospherically significant.

## 6 Conclusions

We have calculated $S_0$ reaction thresholds for nine reaction types in seven classes of carbonyl within a 'small' carbonyl dataset. In general, the $S_0$ transition states are 'late' and resemble the products. Reaction threshold energies typically correlate with the stability of the product molecules, enabling our results to be generalised to larger carbonyls.

In the smallest carbonyls and dicarbonyls, formaldehyde, glyoxal and methylglyoxal, the lowest energy threshold is for direct decarbonylation and this mechanism will compete with NTI dissociation. In larger aldehydes direct decarbonylation via a TS mechanism has a threshold of $\sim$350 kJ/mol in saturated species and $\sim$360–375 in $\alpha,\beta$-unsaturated species. This reaction is therefore likely unimportant in larger aldehydes. However, alternate roaming pathways may be viable if there are low energy barrierless $S_0$ NTI pathways.

In larger carbonyls, the lowest energy $S_0$ dissociation thresholds are for triple fragmentation (TF) of both saturated ($\sim$300 kJ/mol) and $\alpha,\beta$-unsaturated ($\sim$340 kJ/mol) aldehydes, with negligible impact from extension of the main alkyl chain. Branching at the $\alpha$-position only increases TF thresholds by $\sim$5 kJ/mol, but increases the reaction path degeneracy and hence, likely, the QY.

The only 4-centre $H_2$-loss mechanism relevant at actinic energies involves the formyl and $\alpha$ hydrogens, and so need only be considered for aldehydes. This threshold is highest, at 337 kJ/mol, for acetaldehyde, and we expect it to be $\sim$315 in other aldehydes since a terminal double-bond is not being formed.




The NTIII $\beta$-hydrogen transfer reaction is energetically accessible in saturated carbonyls, but largely inaccessible for $\alpha,\beta$-unsaturated species. This pathway is often overlooked in the interpretation of photolysis experiments but should be considered when the production of alkenes and aldehydes shows little-to-no pressure dependence.

We expect alkane elimination to have thresholds of $\sim$350 kJ/mol in linear unsaturated ketones. Whilst energetically accessible under tropospheric conditions, these thresholds are above those for NTI dissociation and we do not expect alkane elimination to be significant. The AE threshold, however, is reduced to $\sim$330 kJ/mol by branching or unsaturation at the $\alpha$ position and alkene elimination may be important in these species.

Finally, where present, tautomerisation pathways have the overall lowest $S_0$ thresholds and yield highly reactive unsaturated species: enols or ketenes. The keto–enol tautomerisation of acetaldehyde to vinyl alcohol leads to significant formation of formic acid in the troposphere. This process may be relevant to other atmospheric carbonyls. In particular, the $\alpha,\beta$-unsaturated carbonyls have UV absorption spectra that extend to low energies, high excited state reaction thresholds and relatively high $S_0$ dissociation thresholds.

The calculations in this paper demonstrate a range of ground state reactions are accessible within the tropospheric "photochemistry" of carbonyls. The energetic thresholds for these $S_0$ reactions are some of the lowest calculated for any carbonyl reaction, on any electronic state. Many of the TF reactions as well as the keto–enol and enal–ketene tautomerisation are predicted to have reaction thresholds $\lesssim$300 kJ/mol. These reactions are likely to be important following photoexcitation at energies below any excited state reaction threshold or following collisional cooling of excited state carbonyl molecules below such thresholds. It may be that the QY for an individual carbonyl is relatively small. The fact that one or more of these reactions are expected to occur in *all* atmospheric carbonyls suggests that, cumulatively, they may be significant in the troposphere and may have atmospheric consequences.

*Supplementary Material:* The supplement related to this article contains a figure and discussion of the energetic thresholds for relevant $S_0$ reactions in butanal; For the reaction classes considered, excepting enal–ketene tautomerisation, a review of the previous computational literature, additional higher energy thresholds, as applicable, and representations of the optimised $S_0$ first-order saddle points (excepting those for $H_2$–loss); Cartesian coordinates for all optimised saddle points are provided as text filed in a ZIP folder. https://doi.org/10.5194/acp-0-1-2021-supplement

*Author contributions.* K.N.R. performed all calculations and data analysis. M.J.T.J. and S.H.K. conceived and directed the project, M.J.T.J. supervised the calculations. All authors contributed to data interpretation and the drafting of the manuscript.

*Competing interests.* no competing interests are present



*Acknowledgements.* This research was undertaken with the assistance of resources and services from the National Computational Infrastructure (NCI), which is supported by the Australian Government, as well as computer time on the computational cluster Katana supported by the Faculty of Science, UNSW Australia, and the computational cluster Artemis supported by the Sydney Informatics Hub at the University of Sydney.

*Financial support.* This work was supported by the Australian Research Council (grants DP160101792 and DP190102013). K.N.R. recognises an Australian Government Research Training Program (RTP) scholarship.



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
