# Peer review of "An assessment of the tropospherically accessible photo-initiated ground state chemistry of carbonyls"

_Atmospheric Chemistry and Physics, 2021_

## Author Comment (AC1)

**RE      MS no: acp-2021-424**

**Title: Photo-initiated ground state chemistry: How important is it in the atmosphere?**

Response to Reviewer's Comments:

The reviewer's comments are indicated as bold text, our response is in normal text and proposed changes to the manuscript are indicated as red text. Line number are with reference to the original manuscript.

Response to RC1

**1. Title, since the entire paper focuses on carbonyls I recommend including that term in the title.**

We have changed the title to:

"An assessment of the tropospherically accessible photo-initiated ground state chemistry of carbonyls"

**2. Table 1, why are a few values given uncertainties but others are not? Do I understand from the footnotes that some reported values in this table are from previous work while the remainder are from the current study? If so I recommend making that distinction more clearly in the table.**

The asymptotic results for Norrish Type I*a* (NT1*a*) and Norrish Type I*b* (NTI*b*) reactions provided with uncertainties in Table 1 are effectively experimental values that have been determined from enthalpies of formation available in the Active Thermochemical Tables. The other NTI*a* and NTI*b* asymptotic energies were previously reported in Rowell et al. (2019). This was originally indicated by a single table footnote in Table 1 for all NTI asymptotic values and a single sentence on line 139 of the text. We recognise this could have been more clearly stated. We have revised the table to include explicit footnotes, either *b* or *c*, for all NTI asymptotic energies (reassigning the other footnote labels), that is:

*a*: NTI*a* and NTI*b* are cleavage to the larger and smaller fragments, respectively;

*b*: Values with uncertainties are determined from experimental enthalpies of formation from the Active Thermochemical Tables (v. 1.22p) (Ruscic et al., 2005; Ruscic, B. and Bross, D. H., 2020); *c*: reproduced from Rowell et al. (2019);  …

We have also expanded the text at line 139 to read:

For some of the carbonyls considered, asymptotic energies for NTI reactions can be obtained from (effectively) experimental enthalpies of formation at 0 K, tabulated in the Active Thermochemical Tables (Ruscic et al., 2005; Ruscic, B. and Bross, D. H., 2020). The results in Table 1 with uncertainties are based on these experimental enthalpies of formation. The remaining NTI asymptotic energies shown in Table 1 have been previously calculated and are reproduced from Rowell et al. (2019).

**3. line 16, "For example, small carbonyls, which rank at the top of anthropogenic emissions". Please be more precise. Do you mean these are the top anthropogenic VOC emissions? The top anthropogenic carbonyl emissions? The top anthropogenic emission of any species?**

We agree the text should have been more precise and have emended the text to read:

For example, small carbonyls such as acetone, formaldehyde and acetaldehyde, which are ranked in the top 25 of all anthropogenically emitted molecules by mass (Simon et al., 2010), are emitted as pollutants (Chen et al., 2014).

**4. line 18, "emitted as biological volatile organic compounds (BVOCs) that oxidise to carbonyls". Note that carbonyls can also be emitted directly by plants.**

We agree and note, in particular, carbonyls are directly emitted by plants, and particularly under heat stress. We have split the original sentence (starting line 16) into two sentences.The sentence now reads:

Up to 10% of carbon initially fixed by plants is also subsequently emitted as biological volatile organic compounds (BVOCs), which include directly emitted carbonyls as well as other volatile species that are subsequently oxidised to carbonyls (Seco et al., 2007).

**5. line 22, "As there are…" this sentence is somewhat awkward and difficult to parse, consider rephrasing**

This sentence has been rephrased to read:

There are of the order of $10^6$ known or suspected VOCs in the atmosphere (Goldstein and Galbally, 2007). As a category, BVOCs are structurally complex and carbonyls formed by their subsequent oxidation will reflect this structural diversity. Consequently, atmospheric carbonyls can have varied chemical structures and reactivities (Kawamura et al., 2000; Atkinson et al. 2006).

**5. line 44, I believe the citation here should refer to the GEOS-Chem model, not to the Harvard atmospheric chemistry group.**

We had cited the Harvard atmospheric chemistry group because we wanted to refer to GEOS-Chem as a model, rather than as a particular version of the GEOS-Chem model. We have now emended this reference to (Bey et al., 2001), which is the first description of the GEOS-Chem model.

**7. Section 1.1. This is a great description and explanation of the processes following UV absorption, well done.**

Thank you so much for the kind words, our goal was to make this as relevant and accessible as possible to the atmospheric chemistry and physics community.

**8. line 162, Figure referenced here should be S3**

The reference was to Section S2, we have made this clearer by specifically referencing Table S1 and Figure S3 in the Supplementary Material as:

(Table S1 and Fig. S3 in Sect. S2 of Supplementary Material)

**9. line 180, reactant here should include R (or R') to match the product**

Because it is the $\beta$ H that forms the $H_2$ molecule, we think it is clearer to give a specific example and have changed this to:

For example, the TF reaction in the saturated aldehyde, propanal is: $CH_3–CH_2–(C=O)H \square H_2C=CH_2 + H_2 + CO$.

**10. line 383, "Photo-initiated keto–enol tautomerisation has only been observed in acetaldehyde (Clubb et al., 2012; Shaw et al., 2018)". While this section of the paper is specific to saturated aldehydes, I think this statement could create confusion and should be clarified – i.e., acetaldehyde is the only saturated aldehyde in which this has been observed. As pointed out later it has also been observed in acetone and MVK.**

We agree this is confusing and have clarified this to read:

Although experimentally observed in ketones, the only aldehyde for which photo-initiated keto-enol tautomerisation has been observed is acetaldehyde (Clubb et al. 2012, Shaw et al. 2018), where TF is not available.

**11. line 464, "This suggests…" I find the wording in this sentence unclear, consider clarifying**

We agree it is unclear and have clarified this sentence to read:

Because of the efficiency of collisional cooling, the most important $S_0$ dissociations are likely to be those with the lowest energetic thresholds, that is, TF if available, NTI in ketones and selected AE, NTIII and formyl+a $H_2$-loss reactions.

**12. line 522, "Molecular hydrogen is an important atmospheric reducing agent", please specify what you mean here**

This has been clarified to read:

Molecular hydrogen is an important atmospheric reducing agent, for example, it reacts with OH to form $H_2O$. Molecular hydrogen is also an indirect greenhouse gas because it increases the atmospheric lifetime of $CH_4$...

Response to RC2

**General comment:**

**The only significant comment I have on this paper is that, while the authors show that ground state chemistry can occur and may be important, they can not show how important it is. With only energetic data available, it can only be determined that a reaction is accessible, but without an assessment of the reaction entropy the rate of the chemical reaction can not be determined and hence no estimate of the yields can be made. Reactions of highly excited intermediates in the ground state through high-energy exit channels has been proposed several times already, but most of the time appears to be of very minor importance as the energy-specific rate coefficients are typically very low. Specifically, the highly excited intermediate has a high state density at those internal energies, while the transition state has a high barrier and hence a low amount of excess energy, leading to a low state density and thus low reaction rates. The threshold argumentation of the authors is therefore only qualitative (a nuanced version of pass/fail), but does not provide any evidence that these reaction can be sufficiently fast to have a non-negligible impact.**

We agree with the reviewer and calculations of rate coefficients and master equation modelling are part of our ongoing research. These points also form the basis of Section 5.1 'Challenges to address' (line 530).

As a benchmarks, we include in Section 5.1 the collision frequency at 1 atm, our previously fit average energy loss per collision of $N_2$ with acetaldehyde and our previous RRKM calculated rate coefficients for the $S_0$ reactions of acetaldehyde, along with the experimental observation of these reactions (lines 542-553 of the discussion). We use these values to show the $S_0$ reactions described in the manuscript are feasible under tropospheric conditions.

**I suggest that the authors more explicitly state that their data does not allow for an assessment of the reaction kinetics and yields, and that their conclusions are thus rather tentative. Alternatively, the authors could use simple RRKM theory with the already available quantum chemical data to calculate approximate energy-specific rates for one or more of the most promising channels and compare those to the collision rate to support their premise. Quantitatively correct master equation studies are not necessary for this paper.**

Addressing this point, and addressing point 6 below, we have expanded the last paragraph of the introduction (line 113) to read:

Ultimately, however, the most important reactions will be those with the highest reaction rate coefficients. Although the threshold energy is, in general, the largest contributor to the magnitude of a reaction rate coefficient, entropy is also important. For similar thresholds, reactions with loose transition states, for example, the variational transition states associated with barrierless reactions like the $S_0$ NTI reactions, will have higher rate coefficients. By identifying reactions that may be important we will be able to focus future work on calculating their reaction rate coefficients and incorporating them into tropospheric master equation models.

We have also emended the final paragraph of the conclusions (line 586) to read:

The calculations in this paper demonstrate a range of ground state reactions are energetically accessible within the tropospheric "photochemistry" of carbonyls. The energetic thresholds for these $S_0$ reactions are some of the lowest calculated for any carbonyl reaction, on any electronic state. Many of the TF reactions as well as the keto–enol and enal–ketene tautomerisation are predicted to have reaction thresholds $\leq 300$ kJ/mol. These reactions are likely to be important following photoexcitation at energies below any excited state reaction threshold or following collisional cooling of excited state carbonyl molecules below such thresholds. An assessment of the likelihood of these $S_0$ reactions and determination of their product yields, however, requires calculation of the respective reaction rate coefficients and master equation modelling. Our results will help target such calculations, as well as future experimental efforts, to reactions most likely to have tropospheric consequences. It may also be that the QY for the $S_0$ reactions of an individual carbonyl is relatively small. The fact that one or more of these reactions are expected to occur in all atmospheric carbonyls suggests that, cumulatively, they may be significant in the troposphere and may have atmospheric consequences.

**Specific comments:**

**1. The title overstates the scope of the paper somewhat, as it is not shown how important the reactions are, merely that they are accessible.**

Our initial title was deliberately provocative and we have changed it to:

"An assessment of the tropospherically accessible photo-initiated ground state chemistry of carbonyls"

**2. The authors use the word "threshold" throughout, but this is ambiguous and it is not clearly defined what is meant. For an endothermic reaction, the threshold energy is the product energy, as tunneling can allow reaction below the TS barrier. The authors seem to refer mostly (but not always?) to the barrier height as the threshold.**

In terms of our quantum chemical calculations, we have used threshold to indicate the limiting zero-point corrected electronic energy for the reaction. For experiment, the threshold is the lowest energy for which reaction has been observed. We have explicitly defined this usage of threshold at line 92 of the original text:

Quantum chemistry methods are used to calculate reaction thresholds for up to nine $S_0$ unimolecular reactions that may be accessible under tropospheric conditions. In our quantum chemistry calculations, the threshold is defined as the limiting zero-point corrected electronic energy for the reaction to occur. Experimental thresholds are the lowest energies for which reaction has been observed to occur and hence include tunnelling through any barrier to reaction. In the figures below, the nine possible unimolecular reactions are colour-coded according to Scheme 3.

**3. p. 2, line 25 "...carbonyls... are one of the few classes of VOC that can absorb solar radiation in the troposphere". Many VOC can, but they are very poor at it (and/or have no reaction pathways in that energy range). I propose adding "efficiently".**

We agree and this has been added.

**4. figure 3: "carbohydrate" is perhaps not the best descriptor of this class. "Hydroxy-aldehyde", supplementing the "linear aldehyde" and "branched aldehyde" classes might be more descriptive.**

We agree. This is an excellent suggestion and we have made this change in Figure 3, Table 1 and in the Supplementary Material.

**5. p. 5, line 106: "Although these isomers are theoretically accessible at actinic energies, their formation barriers are very high with low barriers for the reverse isomerisation."**

This needs one or more references.

(Heazlewood et al. 2011), referenced in the subsequent sentence, contains the energetic barriers for isomerisation to oxirane and methylhydroxycarbene. The reference has been moved to the end of this sentence.

**6. p. 6, line 113: "The calculated S0 thresholds are used to determine general trends..."**

**Threshold energies (here apparently used as product energies) are only one aspect in the contribution of a channel. I suggest the authors also discuss entropy (or TS rigidity), and TS barrier height.**

We use threshold as defined above, that is, as the limiting zero-point corrected electronic energy for the reaction to occur. Only for NTI reactions, where there is no experimental estimate, does threshold refer to an asymptotic energy. We hope that, with the explicit definition included at line 92, this is unambiguous. We agree with the comment about the role of entropy in determination of the relevant reaction rate coefficients and have emended and qualified the original sentence:

The calculated $S_0$ thresholds are used to determine general energetic trends that can be applied to larger carbonyls and  identify  likely $S_0$ reactions for each class of carbonyl under tropospheric conditions. These reactions are then assessed in terms of their tropospheric significance. Ultimately, however, the most important reactions will be those with the highest reaction rate coefficients. Although the threshold energy is, in general, the largest contributor to the magnitude of a reaction rate coefficient, entropy is also important. For similar thresholds, reactions with loose transition states, for example, the variational transition states associated with barrierless reactions like the $S_0$ NTI reactions, will have higher rate coefficients. By identifying reactions that may be important we will be able to focus future work on calculating their reaction rate coefficients and incorporating them into tropospheric master equation models.

**7. p. 11, line 230: "All but the smallest decrease in threshold are outside the likely accuracy of the B2GP-PLYP-D3 calculations." This may be interpreted to say the opposite of what is meant? Rephrase to say that the energy differences exceed the expected uncertainty on the calculations, except for the smallest 2 kJ/mol value.**

We agree this is confusing and have emended it as suggested:

These decreases in threshold energy exceed the expected uncertainty of our B2GP-PLYP-D3 calculations, except for the smallest 2 kJ/mol energy difference.

**8. p. 16, line 295: It may be worthwhile to mention that the keto-enol tautomerization is greatly enhanced by catalysis by acids and other mobile-H compounds.**

We agree and have added:

… and the authors suggest it may occur in many other carbonyls under tropospheric conditions. In solution or at higher gas pressures, keto-enol isomerisation is also readily catalysed by species containing acidic hydrogens – including water, alcohols, and organic and inorganic acids.

**9. p. 18, line 334: "... the threshold... is ~13 kJ/mol lower...". After rounding that should be 14 kJ/mol. Also, the So et al. value of 298.7 kJ/mol is not significant to 4 digits, and representing that value as "299 kJ/mol" here could make sense.**

We have made these changes – thank you for noticing our rounding error.

**10. p. 19, line 352: "Triple fragmentation has been observed as a primary photolysis mechanism in propanal and 2-methylpropanal, with QYs of 4% and 9%, respectively, at 1 atm pressure of $N_2$" (and similar statements elsewhere).**

**The observation of low yields of the products does not imply that these products are formed from S0 chemistry, and the authors do not state that the literature has unequivocally documented that these products are not formed from the S1/T1/... excited states. This section needs to state explicitly for each observation cited that it is known/shown that the products are from S0, or that it is not (yet) clear that S0 has a significant contribution in these observations.**

This is a pertinent observation. Experimentally, we can identify the products as primary products on the basis of the pressure dependence of their quantum yields or on additional experiments, for example, single molecule velocity map ion-imaging experiments. Our assertion that these experimentally observed mechanisms occur on $S_0$ is based on thermal experiments, where excited state reaction is not possible, kinetic modelling and/or the absence of an accessible excited state dissociation pathway. We have emended the text as follows:

Line 352: Triple fragmentation has been observed as a primary photolysis mechanism in propanal and 2-methylpropanal, with QYs of 4% and 9%, respectively, at 1 atm pressure of $N_2$ (Kharazmi, 2019). On the basis of kinetic modelling and, in the absence of excited state primary pathways, both sets of TF products were found to be consistent with $S_0$ reaction. The increased QY for 2-methylpropanal over propanal was attributed to …

Line 359: This channel has been observed experimentally, with a QY of ~1% at 305 nm (392 kJ/mol) and 1 atm of $N_2$ and is demonstrated to be a primary process by single molecule velocity map ion-imaging (VMI) experiments (Harrison et al., 2019). Although classical trajectory simulations initiated at the $H_2$-loss TS could not reproduce the VMI product state distributions, they are consistent with an $S_0$ roaming mechanism (Harrison et al., 2019). The equivalent $H_2$-loss thresholds for the other saturated aldehydes …

Line 363: Indeed, under tropospheric conditions, methylketene and dimethylketene have been observed as minor products following photolysis of propanal and 2-methylpropanal, respectively, with kinetic modelling consistent with $S_0$ reaction (Kharazmi, 2019).

Line 378: … and ~0.2% for 2-methylbutanal (Gruver and Calvert, 1958). Although two $S_0$ decarbonylation mechanisms are known, roaming reactions and reaction via a TS, there is no known excited state decarbonylaiton mechanism. Given the relatively high threshold energies, we predict other $S_0$ pathways will be favoured in larger saturated aldehydes over decarbonylation…

Line 384: … keto-enol tautomerisation has been observed is acetaldehyde (Clubb et al., 2012; Shaw et al., 2018), where TF is not available and master equation modelling confirms that it is an $S_0$ process (Shaw et al., 2018).

Line 395: … experimental examples of the, assumed $S_0$, NTIII mechanism in the literature.

Line 397 … keto--enol tautomerisation in acetone under thermal conditions, where reaction must occur on $S_0$ (Couch et al., 2021), suggests …

Line 421: … Tautomerisation has been observed in methyl vinyl ketone under thermal conditions (Couch et al., 2021), and $S_0$ tautomerisation may therefore …

**11. p. 20, line 383: "Dissociation of the aldehyde reduces its concentration and hence reduces the rate of enol formation."**

**It reduces the yield of enols, but the rate (rate coefficients) remains the same.**

The rate is the product of the concentration and the rate coefficient and, while the rate coefficient is indeed constant, the overall rate will reduce as concentration reduces. To avoid confusion, we have emended the sentence to read:

Dissociation of the aldehyde reduces its concentration and hence, although the rate coefficient is constant, the rate of enol formation is reduced.

**12. p. 22, line 467-471**

**Fast reversible keto-enol isomerisation increases the accessible state density at the higher energies, and hence lowers the effective rate of the other (higher-energy) channels by decreasing the effective concentration of the carbonyl in favor of a reservoir as enol. It is thus not a given that these channels have no influence on the importance of the other channels.**

This is an excellent point and we agree. We have emended the text to read:

469: Tautomerisation, per se, is therefore unlikely to be important if there are low energy $S_0$ dissociation pathways. It will be important in the absence of such a channel, for example, in acetaldehyde (Clubb et al., 2012; Shaw et al., 2018), some ketones and some $\alpha,\beta$-unsaturated carbonyls. Tautomerisation may also have an indirect effect on other $S_0$ reactions. Rapid interconversion of tautomers will reduce the effective concentration of the 'keto' form and hence reduce the probability of other possible $S_0$ reactions.

**13. p. 23, line 505: "In both polluted and pristine regions there are discrepancies between the predictions of atmospheric models and field measurements of the concentrations of organic acids" For your information: a recent paper by Franco et al. (DOI: 10.1038/s41586-021-03462-x) proposes a pathway that may close the gap.**

We thank the reviewer for this and have included it as an additional reference.

506: and field measurements of the concentrations of organic acids (Cady-Pereira et al., 2014; Yuan et al., 2015; Millet et al., 2015), although a recent paper by Franco et al. (2021) proposes a pathway that may close the gap.

**14. Supporting information: use the more exact terms "product energy" or "reaction barrier" rather than "threshold". If both TS and product energies are available, it could be worthwhile to show both.**

Only the barrierless NTI reactions have threshold energies equal to the product asymptotic energy and these are not considered in the Supplementary Material as they were the subject of a previous paper. We have therefore changed 'threshold' to 'zero-point corrected barrier height' throughout the Supplementary Material.

**Typos:**

**15. Several instances of "formaldehdye", "glycolaldehdye", "crotonaldhyde"**

Fixed.

**16. p. 21, line 433: space between "2CO"**

Done

---

## Author Response (AR2)

**Meredith Jordan**

Professor of Theoretical Chemistry, School of Chemistry

15/11/2021

Dr Andreas Hofzumahaus

Editor, Atmospheric Chemistry and Physics

**RE      MS no: acp-2021-424**

**Title: Photo-initiated ground state chemistry: How important is it in the atmosphere?**

Dear Dr Hofzumahaus,

Responding to your comment:

**In the title and throughout the paper, you are using the term 'carbonyl' for organic carbonyl compounds. In order to avoid possible confusion with inorganic or organometallic carbonyl compounds, at least the title should read '.... organic carbonyl compounds'.**

We have changed the title to "An assessment of the tropospherically accessible photo-initiated ground state chemistry of organic carbonyls". We also explicitly refer to "organic carbonyls" in the first paragraph. The first two paragraphs also explicitly reference volatile organic compounds. We subsequently use the generic term "carbonyl".

Please do not hesitate to contact me if you have any questions.

Sincerely,

Meredith Jordan

(for all authors)

School of Chemistry
Faculty of Science
Room 201A, School of Chemistry, F11
The University of Sydney
NSW 2006 Australia

**T** +61 2 9351 4420
**F** +61 2 9351 3329
**M** +61 439 489 547
**E** meredith.jordan@sydney.edu.au

ABN 15 211 513 464
CRICOS 00026A